# Synchronized LFP rhythmicity in the social brain reflects the context of social encounters

Alok Nath Mohapatra [1✉], David Peles[1], Shai Netser[1] & Shlomo Wagner [1]

Mammalian social behavior is highly context-sensitive. Yet, little is known about the mechanisms that modulate social behavior according to its context. Recent studies have revealed a network of mostly limbic brain regions which regulates social behavior. We hypothesize that coherent theta and gamma rhythms reflect the organization of this network into functional sub-networks in a context-dependent manner. To test this concept, we simultaneously record local field potential (LFP) from multiple social brain regions in adult male mice performing three social discrimination tasks. While LFP rhythmicity across all tasks is dominated by a global internal state, the pattern of theta coherence between the various regions reflect the behavioral task more than other variables. Moreover, Granger causality analysis implicate the ventral dentate gyrus as a main player in coordinating the context-specific rhythmic activity. Thus, our results suggest that the pattern of coordinated rhythmic activity within the network reflects the subject's social context.

[1] Sagol Department of Neurobiology, Faculty of Natural Sciences, University of Haifa, POB. 3338, Haifa 3103301, Israel. ✉email: thinkalok@gmail.com

Mammalian social behavior is highly complex and dynamic, involving multiple types of distinct, sometimes even opposing, interactions between partners. Indeed, the identity of a partner can completely change the nature and trajectory of social actions taken by an individual[1]. In addition to these complexities, social interactions are highly dependent upon the social context. For example, humans will most likely respond differently to a hand placed upon their shoulder from behind if this happens in a frightening context, such as in a dark alley in a foreign city, then if the same contact occurs at a cocktail party. Such context-dependent hidden processes which determine how brains respond to inputs and produce behavioral outputs are defined as internal states and include arousal, motivation, emotion and varying homeostatic needs[2]. Presently, little is known of the brain mechanisms and neural circuits that encode the context of social encounters and change responses to social cues accordingly.

In the last two decades, studies have begun to reveal the brain circuits that subserve various types of social behavior[3–6]. Such studies exposed the involvement of a vast network of limbic brain regions, here termed the "social brain"[7], in processing social sensory cues and regulating mammalian social behavior[8,9]. These include striatal regions, such as the nucleus accumbens core (AcbC) and shell (AcbSh), the prelimbic (PrL) and infralimbic (IL) prefrontal cortical areas, several hippocampal and septal regions and multiple amygdaloid and hypothalamic nuclei[10–13]. Many of these areas are highly interconnected in a bidirectional manner[14–18], and some were shown to be involved in various, at times opposing, types of social behavior[10,19–22]. It remains, however, unclear how this intricate network of brain areas generates the large repertoire of distinct types of social behavior in a context-dependent manner. Recent studies using multi-site brain recordings from behaving animals have demonstrated that system-level neural activity in sub-networks of the social brain predicts individual social preferences[23], intentions[24] and decision-making[25] better than does local neural activity at any single brain region. These results thus suggest that coding of the complex aspects of social behavior in the brain should be considered at the system level.

Oscillatory neural activity, mostly in the theta (4–12 Hz) and gamma (30–80 Hz) bands, was reported in many cortical and sub-cortical brain regions in various species[26–28], with its power being shown to intensify during demanding cognitive functions, such as learning[29–31] and social communication[32–34]. Moreover, high theta and gamma rhythmicity were associated with various internal states, such as avoidance, fear, anxiety and attention[30,35–37]. Notably, abnormal theta and gamma rhythms have been reported in multiple neurodevelopmental disorders[38–40], such as autism spectrum disorder (ASD)[41,42]. Accordingly, one prominent hypothesis states that coherent manifestation of these rhythms can dynamically coordinate the activity of neural ensembles dispersed over multiple brain regions and bind them into ad hoc functional sub-networks dedicated for specific cognitive and emotional tasks[43–46].

In the present study, we hypothesized that coherent theta and gamma rhythms couple various regions of the social brain into functional sub-networks in a social context-dependent manner. In other words, distinct social contexts dictate different patterns of coordinated rhythmic activity of dispersed social brain neuronal ensembles, which in turn subserve context-dependent processing of social cues and consequent behavioral responses. To test this hypothesis, we recorded extracellular electrical activity simultaneously from multiple regions of the social brain in mice performing three distinct binary social discrimination tasks (social contexts). The same type of stimulus animals (social

stimulus) served as either preferred or less-preferred stimulus in all three tasks. Using this design, we could link distinct patterns of rhythmic neural activity across the social brain to either stimulus identity or its valence, or to the social context. Our results reveal that the pattern of coordinated oscillatory activity (coherence) in the network is strongly correlated with the social context and carries information that may be used to discriminate between distinct, albeit similar, social contexts. Further, we revealed that the ventral dentate gyrus (vDG), an area previously linked to contextual information[47,48], seems to be involved in coordinating the coherent activity among the various regions of the social brain.

## Results

**Analyzing the behavior of CD1 mice during three distinct binary social discrimination tasks.** Using custom-built electrode arrays (EAr) we simultaneously acquired local field potential (LFP) signals from up to 16 brain regions at a time (cumulative count: 18 regions; Supplementary Fig. 1a and Supplementary Data 2) during interactions of adult male mice (subjects; $n = 14$) with various stimuli[49]. We aimed to sample widespread social-behavior associated regions in the neocortex (prefrontal and piriform cortices), striatum (nucleus accumbens and ventral pallidum (VP)), ventral hippocampal formation (e.g. dentate gyrus (vDG) and vCA1), latera septum (LS), amygdala (e.g. basolateral (BLA) and medial (MeAD)) and hypothalamus (e.g. dorsomedial (DMD) and paraventricular (PVN) nuclei). The location of each electrode was verified post-mortem[49], and since the targeting accuracy was limited, not all brain regions were recorded in each subject (see Supplementary Fig. 1a and Supplementary Data 2 for details). For social contexts, we employed three distinct binary social discrimination tasks[50,51], namely the social preference (SP) (Fig. 1a), emotional-state preference (EsP) (Fig. 1e) and sex preference (SxP) (Fig. 1i) tasks[50]. (See timeline in Supplementary Fig. 1b). Each task comprised a five min-long baseline period, involving empty chambers located at opposing corners of the arena, followed by a five min-long encounter period, when a distinct stimulus was introduced into each chamber[52]. Mice performing the SP task tended to interact with stimulus animals for significantly more time than with objects throughout the encounter period (Fig. 1b–d). Similarly, mice performing the EsP task preferred to interact with socially isolated rather than group-housed stimulus animals (Fig. 1e–h), while mice performing the SxP task tended to interact more with female than with male stimulus animals (Fig. 1i–l). Thus, in each task, the subjects discriminated between a preferred and a less-preferred stimulus. Importantly, the same type of stimulus animal (a group-housed male mouse) that was the preferred stimulus in the SP task was the less-preferred stimulus in the other two tasks. Therefore, this set of tasks allowed us to analyze brain-wide neural activity patterns in association with either the type of stimulus (i.e., a group-housed male vs. an object/group-housed female/isolated male) or its valence (i.e., preferred vs. less-preferred), or the social context (i.e., SP, EsP or SxP task). It should be noted, that in our hands ICR female mice do not discriminate between group-housed and isolated animals[53], hence we conducted this study using male subjects only.

We further compared multiple behavioral parameters between the various tasks. There were no significant differences between the tasks in terms of total time dedicated to stimuli investigation (Fig. 1m), the total number of transitions made by the subjects between the two stimuli (Fig. 1n) or the distance traveled by the subjects during a task (Fig. 1o). Nonetheless, the preference (reflected by the relative discrimination index, RDI) between the

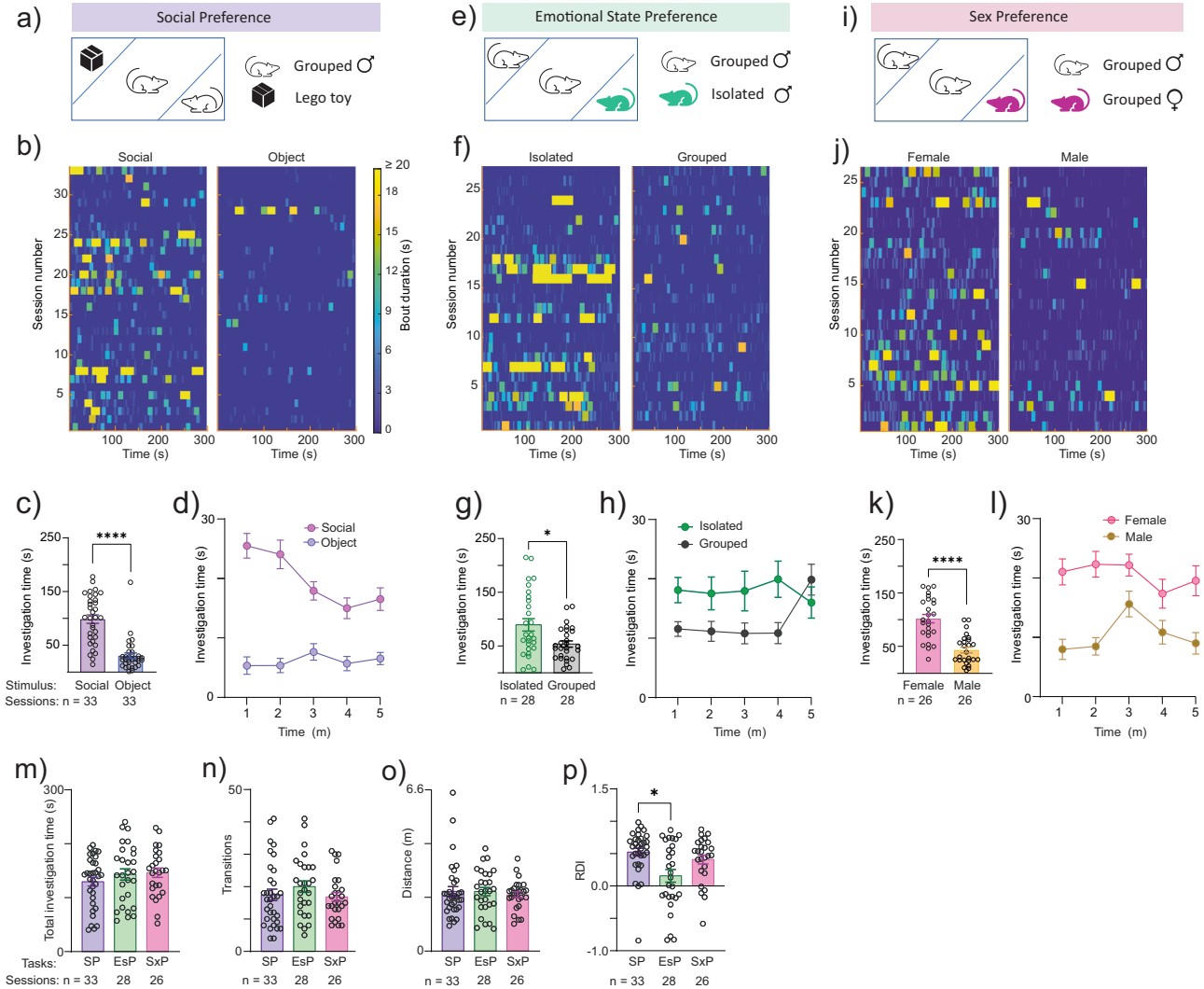

**Fig. 1 Similar behavior of subject mice across three binary social discrimination tasks. a** A scheme of the arena during SP task session. The two types of stimuli are indicated on the right side. **b** Heat maps of investigation bouts made by the subjects toward each of the stimuli (stimulus type is noted above) across the five min-long encounter period of the SP task, with color-coding of the investigation bout duration (see scale on the right of the panel). Each line represents a distinct session. **c** Mean ( ± SEM) time dedicated by a subject for investigating each stimulus during the SP task sessions shown in (**b**). Sample type (sessions) and size ($n =$ ) are denoted below each bar. Wilcoxon matched pairs signed rank test, $n = 33$ sessions, $W = -495$, ****$P < 0.0001$. **d** As in (**c**), plotted vs. time using one min bins. **e–h** As in (**a–d**), for the EsP task. Paired $t$ test, $n = 28$ sessions, $t(27) = 2.374$, *$P = 0.025$. **i–l** As in (**a–d**), for the SxP task. Paired $t$ test, $n = 26$ sessions, $t(25) = 5.75$, ****$P < 0.0001$. **m** Mean (±SEM) total time dedicated by a subject to investigate both stimuli during the encounter stage of each task. Kruskal–Wallis test, $n = 3$ tasks, $H = 1.702$, $P = 0.427$. **n** Mean (±SEM) number of transitions between stimuli made by the subject during the encounter period of each task. Kruskal–Wallis test, $n = 3$ tasks, $H = 2.133$, $P = 0.3442$. **o** Mean (±SEM) distance traveled by the subjects during the encounter stage of each task. Kruskal–Wallis test, $n = 3$ tasks, $H = 0.6782$, $P = 0.776$. **p** Mean (±SEM) relative discrimination index (RDI) for each task. Kruskal–Wallis test, $n = 3$ tasks, $H = 8.509$, $p = 0.0142$; Dunn's post-hoc test, *$P < 0.03$.

two stimuli was lower in the EsP task, as compared to the SP task (Fig. 1p). Overall, subject behavior was similar across the various tasks.

**Different tasks elicit different profiles of rhythmic LFP signals in multiple brain regions.** The power spectral density (PSD) profiles of LFP signals recorded during the encounter period (Fig. 2a, b), qualitatively differed among the various tasks performed by the same subject in a brain region-specific manner (Supplementary Fig. 1c–e). For quantitative comparison, we calculated the mean theta (θP) and gamma (γP) power separately for the baseline and encounter periods of each task, for each brain region. While the mean power during baseline across all regions did not significantly differ between the tasks (Fig. 2c, d), the change in power during the encounter (compared to baseline) for

both theta (ΔθP) and gamma (ΔγP) rhythms was significantly higher for the SP task, compared to the other two (Fig. 2e, f). Notably, no difference was found in either ΔθP or ΔγP between the first and second session of the SP task (Fig. 2g, h), suggesting the power change is task-specific. Thus, despite the similar behavior exhibited by subjects across the tasks (Fig. 1), their system-level brain LFP signals significantly and consistently differed in power across tasks. Specifically, despite involving only one social stimulus, the SP task induced the strongest LFP rhythmicity. When considering each brain region separately, we found that in almost all cases, the mean power of both rhythms was enhanced during the encounter period, as compared to baseline. Notably, the mean power change differed significantly among the various tasks and specific regions (Fig. 2i, j). The similar patterns of ΔθP and ΔγP across the various regions

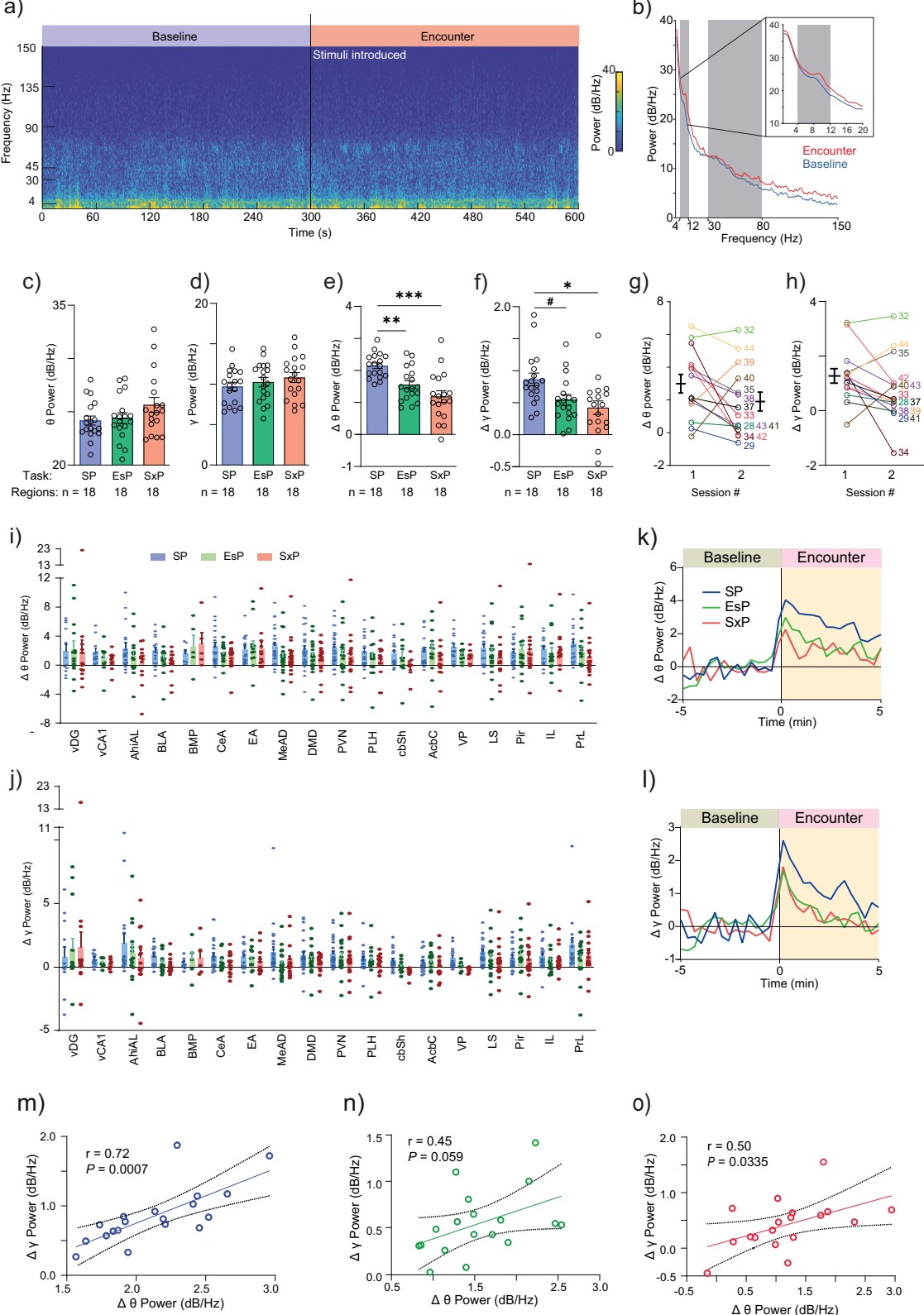

suggested the existence of a link between them. Accordingly, we found statistically significant correlation (Pearson's, $r > 0.25$, $P < 0.05$) between $\Delta\theta P$ and $\Delta\gamma P$ for the SP and SxP tasks, while borderline significant correlation ($r = 0.45$, $P = 0.059$) was observed for the EsP task (Fig. 2m–o). Thus, when measured over the course of the entire session, both rhythms seem to be driven by the same process.

To examine the temporal dynamics of LFP rhythmicity during the various tasks, we plotted $\Delta\theta P$ and $\Delta\gamma P$ as a function of time for each task and brain region. In accordance with our previous study in rats[34], we found that both $\Delta\theta P$ and $\Delta\gamma P$ began to rise several seconds before stimulus introduction, peaked within 20 s from this point and gradually declined in all brain regions and tasks (Supplementary Fig. 2). Thus, the dynamics of LFP

**Fig. 2 Brain region- and context-specific changes in the levels of theta and gamma power during a social encounter. a** Color-coded spectrogram of LFP signals recorded in the AcbSh during the baseline (left) and encounter (right) periods of a session of SP task conducted by a subject. The black line at 300 s represents the time of stimuli introduction into the arena. The color-coding scale is shown on the right. **b** Average PSD profiles of the baseline (blue) and encounter (red) periods from AcbSh from all session of SP Task. The gray areas mark the theta and gamma ranges. The inset shows the theta range in higher resolution. **c** Mean (±SEM) theta power (θP) during the baseline period for each brain region in the three contexts Kruskal–Wallis test, $n = 3$ tasks, 18 areas, $H = 2.725$, $P = 0.2561$. **d** As in (**c**), for gamma power ($\gamma$P; $n = 3$ tasks, 18 areas, Welch's ANOVA, W (DFn, DFd) = 0.9496 (2,33.84), $P = 0.2561$). **e** Mean (±SEM) $\Delta$θP, averaged across all brain regions, for each task. $N = 3$ tasks, 18 areas, W (DFn, DFd) = 14.67 (2, 31.80), $P < 0.0001$. Dunnett's T3 multiple comparisons test, SP vs. EsP, $P = 0.0018$; SP vs. SxP, $P = 0.0002$; EsP vs. SxP, $P = 0.2653$. **f** As in (**g**), for $\Delta$$\gamma$P. $N = 3$ tasks, 18 areas, W (DFn, DFd) = 5.134 (2, 33.65), $P = 0.0113$. SP vs. EsP, $P = 0.0531$; SP vs. SxP, $P = 0.0127$; EsP vs. SxP, $P = 0.7467$. **g** Mean (±SEM, black lines) change in theta power ($\Delta$θP) during the encounter period of the SP task, plotted separately for the first (left) and second (right) session conducted by each subject mouse (mouse number is shown to the right). Lines link the results of each mouse in both sessions. Paired $t$ test: $n = 14$ sessions, $t(13) = 1.722$, $P = 0.1087$. **h** As in (**g**), for change in gamma power ($\Delta$$\gamma$P). Paired $t$ test: $n = 14$ sessions, $t(13) = 1.577$, $P = 0.1388$. **i** Mean (±SEM) change in theta power ($\Delta$θP) during the encounter period, relative to the baseline period for each brain region in the three contexts (2-way ANOVA. Contexts: $F(2, 659) = 3.838$, $P = 0.0220$; Brain regions: $F(17, 659) = 1.727$, $P = 0.0341$; Interaction: $F(34, 659) = 0.4548$, $P = 0.9970$). **j** As in (**i**), for change in gamma power ($\Delta$$\gamma$P; 2-way ANOVA. Contexts: $F(2, 659) = 1.459$, $P = 0.2333$; Brain regions: $F(17, 659) = 2.050$, $P = 0.0076$; Interaction: $F(34, 659) = 0.5732$, $P = 0.9764$). **k** Super-imposed traces of $\Delta$θP averaged across all brain regions for the SP (blue), EsP (green) and SxP (red) tasks. Time '0' represents the time of stimuli insertion. **l** As in (**k**), for $\Delta$$\gamma$P. **m** Mean $\Delta$$\gamma$P as a function of mean $\Delta$θP during the SP task, for each brain region (18 in total). Pearson's correlation coefficient ($r$) and significance ($P$) are given. **n** As in (**m**), for the EsP task. **o** As in (**m**), for the SxP task. Brain region abbreviations: AcbC: Accumbens nucleus, core; AcbSh: Accumbens nucleus, shell; AhiAL: Amygdalo-hippocampal area, anterolateral part; BLA: Basolateral amygdaloid nucleus; BMP: Basomedial amygdaloid nucleus, posterior part; DMD: Dorsomedial hypothalamic nucleus, dorsal part; EA: Extended amygdala; IL: Infralimbic prefrontal cortex; LS: Lateral septum; MeAD: Medial amygdaloid nucleus, anterodorsal; Pir: Piriform cortex; PLH: Peduncular part of the lateral hypothalamus; PrL: Prelimbic prefrontal cortex; PVN: Paraventricular hypothalamic nucleus; vCA1: Field CA1 of the hippocampus, ventral part; vDG: Dentate gyrus, ventral part; VP: Ventral pallidum. #$P = 0.053$, *$P < 0.05$, **$P < 0.01$, ***$P < 0.001$. See also Supplementary Figs. 1, 2.

rhythmicity across the session were similar among the various tasks (Fig. 2k–l) and did not seem to reflect the behavioral dynamics (shown in Fig. 1d, h, l). We also found no significant correlation (Pearson's, $P > 0.05$) between the mean power change and speed of the subject during any task for either $\Delta$θP or $\Delta$$\gamma$P (Supplementary Fig. 3a–f).

Overall, the results of power analysis across the encounter period suggest that theta and gamma rhythmicity are driven by an encounter-induced global brain state[2] that shows similar temporal dynamics across tasks, independent of the behavioral dynamics.

**LFP power changes during stimulus investigation bouts are differentially modulated across brain regions and tasks.** Despite the uniform dynamics of LFP rhythmicity during the encounter across all tasks, it may be differentially modulated during specific behavioral events, such as stimulus investigation. We thus examined the possibility that during investigation bouts, $\Delta$θP and $\Delta$$\gamma$P (henceforth termed $\Delta$θP and $\Delta$$\gamma$P) differ between the various stimuli and tasks. For this analysis, as well as for all other analyses of investigation bouts in this study, we used only bouts which are longer than 2 s, as no significant difference between the stimuli was found for shorter bouts (Supplementary Fig 1f) and since theta coherence cannot be reliably calculated for shorter bouts. As exemplified by signals recorded from the extended amygdala (EA) shown in Fig. 3a–f, a Z-score analysis revealed elevation in gamma power during investigation bouts made towards social but not object stimuli in the SP task, towards grouped but not isolated stimuli in the EsP task and towards male but not stimuli female in the SxP task. All these differences between stimuli were statistically significant (Supplementary Fig. 4a–c). This analysis thus suggests a bias in the response towards specific stimuli, in a task-specific manner. Interestingly, this area showed a bias towards the same type of stimulus used in all tasks (a group-housed male), suggesting that at least for the EA, the change in gamma power was dictated by the stimulus type.

To explore the stimulus-specific bias in LFP power change during each task, we calculated the difference in $\Delta$θP and $\Delta$$\gamma$P between the two stimuli, separately for each brain region. Since a possible bias of LFP rhythmicity of a given brain region may be

associated with the behavioral preference towards a specific stimulus, we examined the correlation between the two variables. To this end, we correlated the $\Delta$θP bias (preferred stimulus minus less-preferred stimulus) to the RDI values of each task. We found a negative correlation in a specific set of brain regions (i.e, EA and LS) only for the SP task. In contrast, a positive correlation was found in a distinct set of brain regions (AcbSh, amygdalo-hippocampal area (AhiAl), VP and DMD) during the EsP and SxP tasks. Specifically, the VP exhibited a very strong and highly significant linear correlation with RDI values during both the EsP and SxP tasks (Fig. 3g). These results suggest a link between stimulus-specific bias in $\Delta$θP and behavioral preference in a task-specific manner.

To further explore this link, we plotted $\Delta$θP bias across all tasks on a 3D plot (x-axis: SP, y-axis: EsP and z-axis: SxP), separately for each brain region (see 2D plots in Supplementary Fig. 4g–l). We found that almost no region (with the exception of vDG and vCA1) showed bias towards the object stimulus in the SP task, with the various regions being equally distributed between the two stimuli in the EsP and SxP tasks (Fig. 3h and Supplementary Fig 3a–c). In contrast, when $\Delta$$\gamma$P was analyzed (Fig. 3i and Supplementary Fig 3d–f), we observed an opposite picture. Here, almost all brain regions exhibited a bias towards the grouped and male stimuli in the EsP and SxP tasks, yet were rather equally distributed between the two stimuli in the SP task. Thus, for $\Delta$$\gamma$P, most brain regions (14/18) were equally divided between those biased towards less-preferred stimuli (i.e., object+grouped+male) and those biased towards the type of stimulus used in all three tasks (i.e, social+grouped+male). The probability of such an arrangement to occur by chance is smaller than 0.001 (1-binomial test) for each of these two groups. The results thus suggest that a bias in gamma power is mostly associated with the characteristics (i.e, valence or type) of the stimulus. They also generally demonstrate opposite stimulus-dependent bias patterns between the theta and gamma rhythms during stimulus investigation, in contrast to their significant correlation when measured during the entire encounter period (Fig. 2m–o). This implies the existence of an independent active state in the social brain specifically during stimulus investigation. Notably, of all brain regions considered, the vDG stood out as the only region biased to the combination of

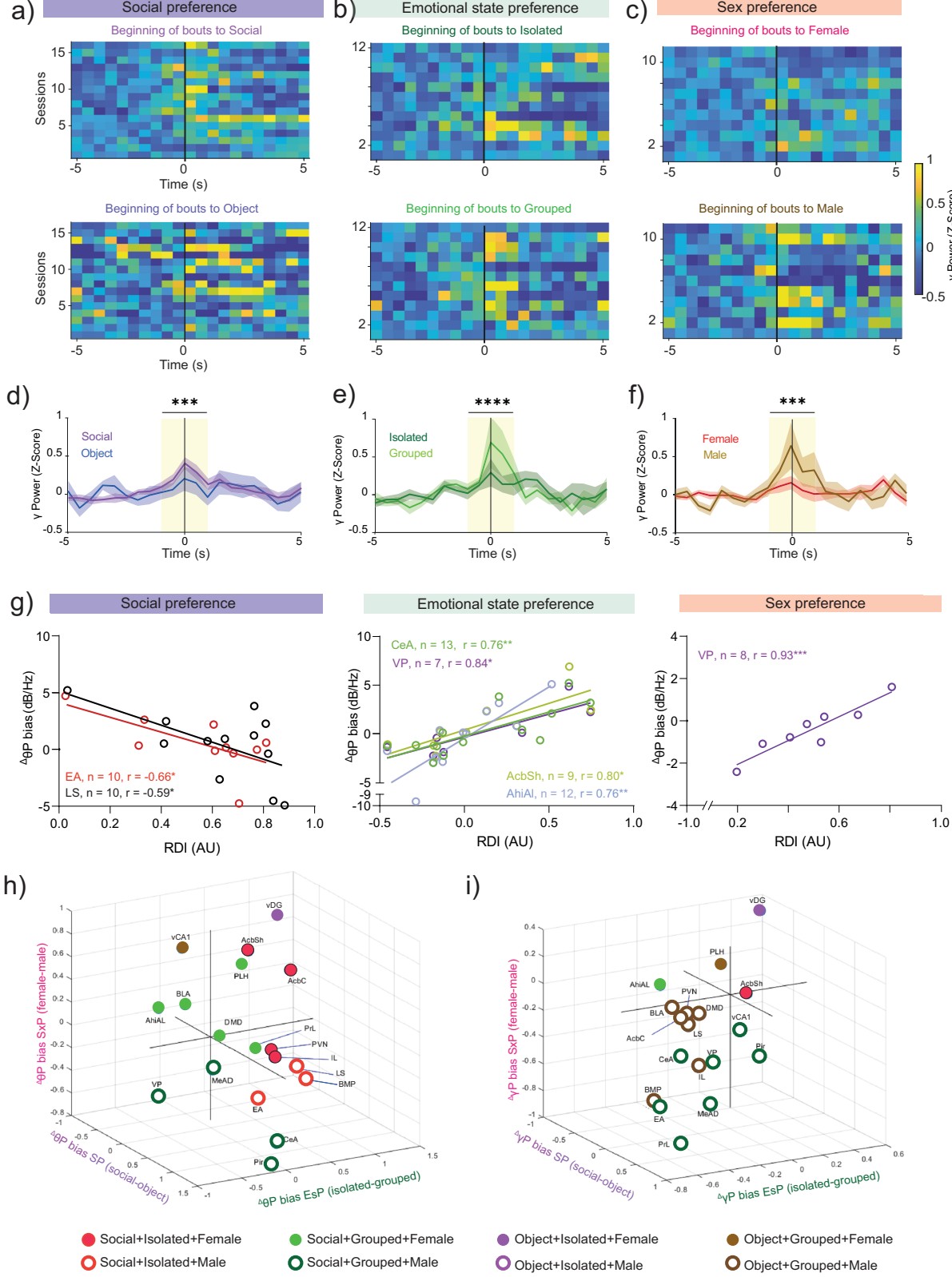

**Social+Isolated+Female** **Social+Grouped+Female** **Object+Isolated+Female** **Object+Grouped+Female**

**Social+Isolated+Male** **Social+Grouped+Male** **Object+Isolated+Male** **Object+Grouped+Male**

object/isolated/female stimuli. Moreover, this region showed an especially strong bias for all these stimuli in both $^\Delta\theta P$ and $^\Delta\gamma P$ (Fig. 3h, i and Supplementary Fig. 4g–l). These results raise the possibility of a unique position for the vDG in the social brain, as also supported by results presented below.

Overall, the analysis of theta and gamma power changes across the tasks revealed a stimulus-specific pattern of changes during active stimulus investigation, on top of an encounter-induced global state eliciting a general increase in theta and gamma power.

**Social encounters modulate coherence between brain regions in a social context-dependent manner.** Synchronous activity (coherence) enhances effective communication between neuronal populations in different brain regions and dynamically binds

**Fig. 3 Stimulus- and task-specific LFP power changes during investigation bouts. a** Heat maps of average change in gamma power of LFP signals recorded in the extended amygdala (EA), before and during social investigation bouts made by the subjects with social (above) and object (below) stimuli during SP task sessions ($n = 17$ sessions). Each row represents the mean Z-score of all bouts in a single session (using 0.5 s bins). Time '0' represents the beginning of the bout. The color code scale is on the right. **b** As in (**a**), for investigation bouts of isolated (above) and grouped (below) social stimuli during EsP task sessions ($n = 14$ sessions). **c** As in (**a**), for investigation bouts of female (above) and male (below) social stimuli during SxP task sessions ($n = 13$ sessions). **d** Mean (±SEM) Z-score trace of the data shown in (**a**) for both stimuli. Yellow bar represents the area where the signal was averaged for statistical comparison between the stimuli. Wilcoxon matched pairs signed rank test, $n = 15$ sessions, $W = -91$, ***$P = 0.002$. **e** As in (**d**). for the data shown in **b**. Paired $t$ test: $n = 12$ sessions, $t(11) = 8.379$, **** $P < 0.0001$. **f** As in (**d**), for the data shown in (**c**). Paired $t$ test: $n = 11$ sessions, $t(10) = 6.099$, *** $P = 0.001$. **g** Correlation between mean change in theta power during investigation bouts ($^\Delta\theta P$) in specific brain regions and RDI values during the various tasks. Only statistically significant linear correlations are shown. Each circle represents $^\Delta\theta P$ and corresponding RDI during a single session. **h** A 3D plot of the mean difference between preferred and less-preferred stimuli in $^\Delta\theta P$. Each circle represents a given brain region, color- and shape-coded according to the combined bias across all tasks. See legend of the color and shape code of the distinct combinations below. **i** As in (**h**), for $^\Delta\gamma P$. See also Supplementary Figs. 3, 4.

them into functional sub-networks[43]. Accordingly, we hypothesized that coherent theta and gamma rhythms couple various regions of the social brain into functional sub-networks in a social context-dependent manner. We, therefore, examined the coherence of LFP rhythmicity between each pair of brain regions recorded by us, in both the theta and gamma bands. During the baseline period, the mean theta coherence ($\theta$Co) across all pairs of brain regions (99 pairs with ≥5 sessions from at least two subjects in all three tasks) was similar across all tasks (Fig. 4a). Thus, the subjects displayed similar global brain synchronization while exploring the arena without stimuli, in all tasks. However, the change in theta coherence ($\Delta\theta$Co) during the encounter period differed significantly between tasks. While almost all pairs of brain regions exhibited increased $\theta$Co during the SP task, we observed significantly milder increases during EsP and SxP tasks, with many paired regions showing reduced $\theta$Co (Fig. 4b, e–g). Comparable relationships among tasks were observed for changes in gamma coherence ($\gamma$Co), although here the general tendency was one of decreased coherence during the encounter period (Fig. 4c–g). Similarly, analysis of the coherence of each brain region with all others revealed a subset of regions which displayed a significantly higher coherence change during the SP task, as compared to at least one task (Supplementary Fig. 5). Thus, the coherence between brain regions showed higher increase during SP encounters, compared to EsP and SxP, at both theta and gamma bands. Notably, while $\theta$Co and $\gamma$Co at baseline showed weak but significant (or borderline significant) correlation with the distance between the brain regions in each couple, no such correlation was observed for the change in theta or gamma coherence ($\Delta\theta$Co, $\Delta\gamma$Co) during the encounter (Supplementary Fig. 4d–g). These results suggest no effect of volume conductance on the coherence change during the encounter. Finally, when calculating the correlations in $\Delta\theta$Co across all paired regions between the various tasks, we found a statistically significant high correlation between SxP and EsP, while no correlation was found between SP and SxP. A milder but significant correlation was found between SP and EsP, whereas all correlations were found significant for $\Delta\gamma$Co (Fig. 4h, i and Supplementary Fig. 6). Altogether, these results suggest a task-specific pattern of changes in theta coherence across brain areas, when this pattern is measured across the whole encounter.

**Brain-wide coherence changes during investigation bouts reflect the social context.** So far, we analyzed the coherence across the whole encounter period, with no consideration of the investigation bouts. To explore possible modulation of LFP coherence specifically during investigation bouts, we calculated the mean $^\Delta\theta$Co between each pair of brain regions during all investigation bouts towards a given stimulus, similarly to how we analyzed the power changes (Fig. 3). Since data had been collected

for a large number of brain-region pairs (99 pairs), we focused on pairs showing a mean coherence change that crossed a cutoff value $\pm 1.5$*standard deviation (SD) for each stimulus (about 20% of the pairs). When plotting the bias in $^\Delta\theta$Co for each of these pairs, separately for each stimulus (Fig. 5a), we found that the three stimuli of the same type (i.e, social, grouped, male) did not share even a single pair of brain regions that passed the $^\Delta\theta$Co cutoff value. Similarly, the preferred stimuli (i.e, social, isolated, female) also did not share even a single pair among them. In contrast, multiple pairs of brain regions shared similar changes in theta coherence between both stimuli used in each task (Fig. 5a). For example, CeA-PrL and MeAD-VP showed increased coherence for both social and object stimuli, BLA-LS and EA-Acbc showed increased coherence for both isolated and grouped stimuli and Pir-AcbC and EA-AcbC exhibited the strongest increase in theta coherence for both male and female stimuli. Thus, changes in coherence during stimulus investigation seem to be dictated by the social context rather than by stimulus characteristics, such as its type or valence.

For quantitative examination of this possibility, we calculated the correlation across all brain regions for either $^\Delta\theta$Co (Fig. 5b) and $^\Delta\gamma$Co (Fig. 5c), between pairs of stimuli which share common context, type or valence. We found strong and highly significant correlations between all pairs of stimuli used in the same task (sharing context). In contrast, among the three stimuli of the same type, only grouped (ESP) and male (SxP) showed a significant correlation. Similarly, among preferred stimuli, only isolated (EsP) and female (SxP) showed significant correlations. Notably, in both of these cases, the correlation was weaker than the correlation between any pair of stimuli sharing the same context (Fig. 5b). Similar results were found for $^\Delta\gamma$Co (Fig. 5c). Thus, coherence changes during stimulus investigation in both bands had the strongest association with the context of the social interaction, relative to any characteristic of the stimulus.

Finally, we employed a Decision trees classifier (multi-class Random forest, see Methods) to examine if $^\Delta\theta$Co and $^\Delta\gamma$Co contain information which may be used to discriminate between the various contexts or stimuli. First, we validated that, when trained over two stimuli, the model achieved good (~60%) and significant accuracy in predicting the social stimulus vs. object in the SP task using either $^\Delta\theta$Co or $^\Delta\gamma$Co (Supplementary Fig. 7a, b). Then, we retrained the same model for predicting the social context among the three options (SP, EsP and SxP), and found that using $^\Delta\theta$Co (Fig. 5d), but not $^\Delta\gamma$Co (Fig. 5e), allowed the model to predict the right context better than any other context, and that this prediction was the only one achieving more than a chance level (33.3%) accuracy (although only SxP classification was statistically significant). In contrast, the same model worked poorly for predicting stimulus identity among all six stimuli (Supplementary Fig. 7c, d). Using the LFP theta power ($^\Delta\theta$P) for

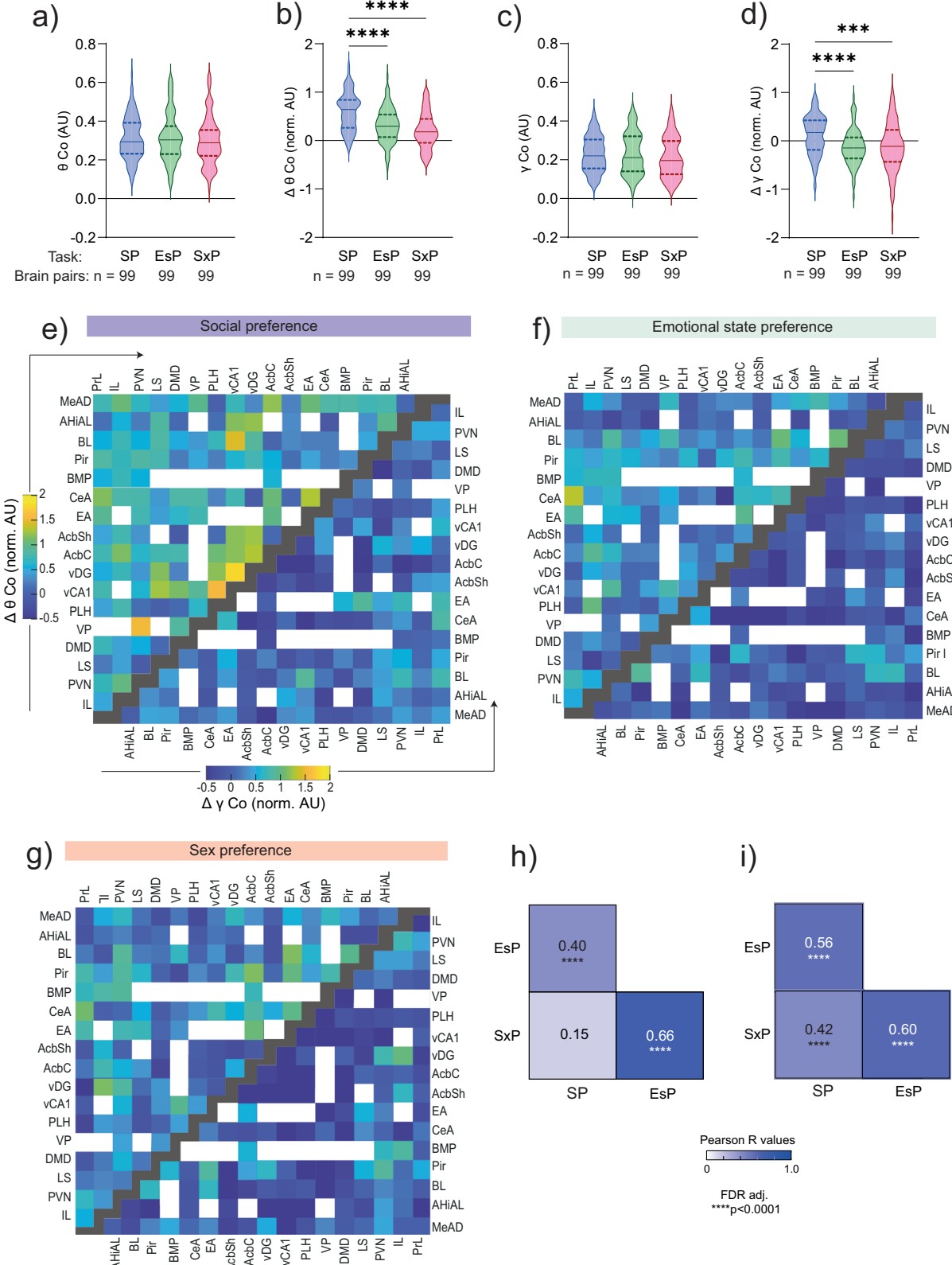

predicting the social context by the same model achieved good and significant classification only for the SP (Supplementary Fig. 7e), while using both theta power and coherence allowed accurate prediction of both SP and SxP contexts (Supplementary Fig. 7f). These results suggest that LFP rhythmicity in the theta range, especially the pattern of coherence between the various brain regions, is informative regarding the social context of the

animal more than regarding the identity or valence of the social stimuli.

**Analysis of Granger causality suggests that specific brain regions serve as hubs.** The coherent LFP rhythmicity in the social brain can be dominated by specific regions serving as hubs, thereby preceding other regions in terms of rhythmic neural

**Fig. 4 Social encounters modulate the coherence between brain regions in a social context-dependent manner. a** Mean theta coherence during the baseline period for each task, across all pairs of brain regions ($n = 99$) recorded during all tasks (Kruskal–Wallis test, $H = 0.75$, $P = 0.687$). The task and sample size are detailed below. **b** As in (**a**), for mean normalized change in theta coherence (ΔθCo) during the encounter period. Note the significant difference between the SP and other tasks (Kruskal–Wallis test, $H = 47.5$, P < 0.0001, Dunn's post-hoc, SP vs. EsP, $Z = 4.723$, ****P < 0.0001, SP vs. SxP, $Z = 6.709$, ****P < 0.0001, ESP vs. SxP, $Z = 1.985$, $P = 0.1414$). **c, d** As in (**a, b**), for gamma coherence (**c**: Kruskal–Wallis test, $H = 2.211$, $P = 0.331$; **d**: Kruskal–Wallis test, $H = 23.58$, P < 0.0001, Dunn's post-hoc, SP vs. EsP, $Z = 4.308$, ****P < 0.0001, SP vs. SxP, $Z = 4.095$, ****P < 0.0001, ESP vs. SxP, $Z = 0.2127$, P > 0.9). **e** Color-coded matrix of the mean normalized ΔθCo (upper left) and ΔγCo (lower right) values for all pairs of brain regions in the SP task. Empty spots represent brain region pairs with fewer than five recorded sessions. Black spots separate between the ΔθCo and ΔγCo matrices. **f** As in (**e**), for EsP. **g** As in (**e**), for SxP. **h** Coefficients and significance of Pearson's correlations of ΔθCo across all coupled brain regions ($n = 99$) for each pair of tasks (****P < 0.0001, FDR corrected). **i** As in (**h**), for ΔγCo. In plots (**a**), (**b**), (**c**), and (**d**), the shaded area between the dashed lines shows the range between the 25th and 75th percentiles, and the solid line within that area represents the median. See also Supplementary Figs. 5, 6.

activity[54]. To identify hub candidate regions, we first selected brain regions which are statistically over-represented (see Methods) among pairs of regions exhibiting strong (mean ± 1.5*SD) bias in any task, separately for ΔθCo and ΔγCo (Fig. 6a, b). We then examined the dependence of LFP rhythmicity of each of these regions in terms of preceding rhythmicity of other regions, by calculating the change in Granger causality (GC)[55] during the encounter period, as compared to the baseline. We found distinct patterns of statistically significant changes in GC (encounter vs. baseline periods) between the various tasks (Fig. 6c–h). Some regions, however, presented significant GC changes in all tasks, suggesting that they might function as hubs. For example, the vDG and AcbC participated in significant GC changes in both theta and gamma rhythms in all tasks. At the same time, PrL and AhiAl were involved in significant theta GC changes in all tasks. Interestingly, theta GC changes from vDG to AhiAl increased during the SP task but decreased during the EsP task, while theta GC changes from PrL to vDG decreased during both EsP and SxP tasks. These results suggest that these brain regions dictate LFP rhythmicity in the social brain during social investigation, in a context-dependent manner.

To further explore this possibility, we have calculated the difference in GC change during encounter between the two directions (from area 1 to area 2 and vice versa), for all couples of brain regions across all tasks and rhythms (Supplementary Fig. 8a–c). After applying FDR correction for multiple comparisons, we found only vDG to LS, for gamma rhythmicity of the EsP task, which was significantly higher in the vDG-LS direction than in the opposite direction (Supplementary Fig. 8d). This result further supports a pivotal role of vDG in coordinating rhythmic neuronal activity in the social brain.

**Context-specific synchronization of LFP rhythmicity in the ventral dentate gyrus with precise behavioral events**. To further examine the candidate hub regions, we exploited our ability to determine the exact timing of each investigation bout, to quantify the synchronized modulation of LFP rhythmicity relative to these events. Thus, we compared the modulation of theta and gamma power in all regions associated with significant GC changes (Fig. 6) relative to a defined battery of specific behavioral events. These events included the beginning and end of investigation bouts towards specific stimuli, as well as transitions between stimuli, as compared to repeated investigation of the same stimulus (Fig. 7a). We found a main effect in ANOVA for multiple events, although in most cases, none of the regions showed significance *in post-hoc* analysis (see Supplementary Data 3). One region, the vDG, did, however, exhibit significant differences between stimuli. The vDG displayed significantly reduced theta and gamma powers at the end of investigation bouts of social stimuli, as compared to object stimuli, specifically in the SP task (Fig. 7b–g and Supplementary Fig. 9a, b). The same region also exhibited decreased theta and gamma powers at the beginning of

transitions from isolated to grouped stimuli, as compared to repeated investigation bouts of grouped stimuli, in the EsP task, as well as increased theta power in the case of transitions from social to object stimuli in the SP task (Fig. 7h–m and Supplementary Fig. 9c, d).

These results, together with those shown in Fig. 3h, i and Fig. 6, suggest that the vDG may function as a hub in the social brain network by coordinating rhythmic neural activity of the network in a social context-dependent manner.

## Discussion

In this study, we used multi-site electrophysiological recordings from the social brain in behaving mice to seek system-level neural correlates of three distinct aspects of social interaction, namely, the type of the social stimulus, its relative valence (preference) and the social context. To distinguish between these three aspects, we relied on three social discrimination tasks (i.e., SP, EsP, and SxP) in which male mice clearly prefer one of two distinct stimuli. This design enabled us to employ the same type of social stimulus, a novel group-housed male mouse, in all three tasks, with this stimulus being the preferred stimulus in the SP task and the less-preferred stimulus in the other two tasks. Importantly, all three tasks took place in the same experimental arena, which enables uniform interactions between the subject and the stimuli, i.e., stimulus investigation by the subject[52]. Consequently, as much as we could measure, subject behavior was almost identical in all three tasks. Therefore, behavioral differences cannot explain the significant differences in the patterns of rhythmic LFP signals observed among the different tasks.

We analyzed LFP signals at three different time resolutions: across the whole session, during stimulus investigation bouts, and in alignment with specific behavioral events, such as at the beginning and end of specific investigation bouts. When analyzing the power of both theta and gamma rhythms over an entire session, some aspects seemed to be dictated by a global internal state, which affected all brain regions. In accordance with previous studies by us and others[33,34], virtually all brain regions exhibited higher levels of theta and gamma power during encounter, as compared to the baseline period. Our observation that the level of enhanced power was both brain region- and task-specific strongly suggests that the power elevation was not caused by enhanced electrical noise or any other artifact but rather by a genuine internal state of the animal. The uniform dynamics of both theta and gamma power changes across all brain regions and tasks during the encounter period further support the existence of a global internal state which is independent of behavioral dynamics or context. Notably, we observed significant correlations across brain regions between theta and gamma power changes in all tasks, suggesting that both rhythms are similarly influenced by the global internal state. In agreement with our previous studies in both rats and mice[32,34], theta and gamma power maintained their high levels for a time, even after the

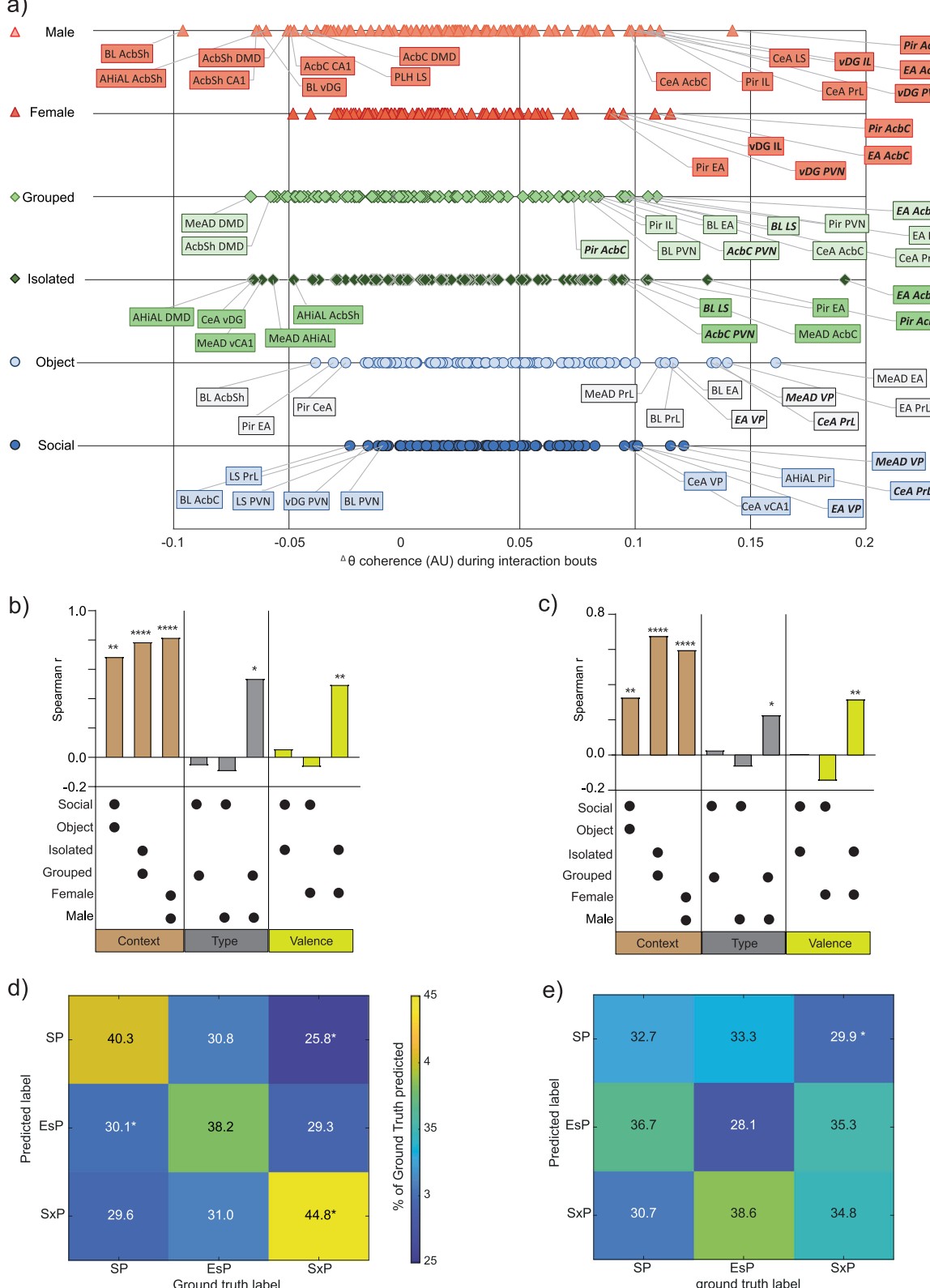

removal of the stimuli from the arena, further supporting an encounter-induced global internal state, which slowly fades away. This state did not seem to be caused by subject movement, as we found no correlation between subject speed and changes in theta or gamma power for any brain region. Whether this internal state reflects arousal, aggression, social motivation or other emotions, is yet to be determined by future studies.

While the dynamics of the internal state seemed to be similar across the distinct contexts, other aspects of the general (session-wise) changes in theta and gamma power exhibited context-specific characteristics. For example, the general changes in both power and coherence were highest in the SP task, suggesting a higher level of the internal state inducing them. Assuming that the global internal state dominating the social encounter reflects

**Fig. 5 Coherence changes during social investigation are informative regarding the social context. a** Distribution of changes in theta coherence during investigation bouts ($^\Delta\theta$Co) between each pair of brain regions, plotted separately for each stimulus used in the SP (blue), EsP (green) and SxP (red) tasks. The names of brain region pairs which passed the mean cutoff value ± 1.5*SD are labeled, with those showing similarly high $^\Delta\theta$Co values for both stimuli of the same task in bold. **b** Spearman's correlation coefficients of mean $^\Delta\theta$Co across all paired brain regions ($n = 99$), for couples of stimuli which were either used in the same task (left, brown bars), of the same type (middle, gray) or having the same valence (right, yellow). The correlated two stimuli are denoted by asterisk below each bar, while the statistical significance of the correlation is marked above the bars (*$P < 0.05$, **$P < 0.01$, ***$P = 0.001$, ****$P < 0.0001$, FDR corrected). **c** As in (**b**), for $^\Delta\gamma$Co. **d** A color-coded confusion matrix for a multi-class Random forest classifier employed for predicting the social context from $^\Delta\theta$Co values across all brain regions and stimuli. The scale of the accuracy's color code is shown to the right. The percentage of cases in a label was predicted for each ground truth are marked in the middle of each spot. *$P < 0.05$, Mann–Whitney test, FDR corrected. **e** As in (**d**), for $^\Delta\gamma$Co. See also Supplementary Fig. 7.

mainly social motivation, these results are somewhat surprising, given how the SP task involved only one social stimulus and reasoning that among the various stimuli tested, the female would be the most attractive to the male subjects. Our interpretation is that the SP task is associated with a higher level of social motivation, as it requires the animal to choose between an inanimate object and a conspecific, while the other two tasks involve two social stimuli, thereby presenting the subject with a more challenging dilemma. The higher motivation of the subject during the SP task is in accord with the simpler pattern of theta coherence changes observed during this task (seen as a general increase across almost all brain region pairs). Overall, these results suggest that the internal state level may distinguish between some contexts, which is in accordance with the ability of the Random forest model to predict only the SP context based on the LFP power. Nevertheless, the changes in theta and gamma power across the encounter period did not differ between the EsP and SxP tasks, and thus cannot be the sole basis for the context-specific responses to social cues.

Analysis of the power change specifically during stimulus investigation, yielded a different picture than did session-wide analysis. First, we found no correlation between theta and gamma power changes during these periods, suggesting a distinct state of active sensing which characterizes stimulus investigation. Moreover, although both theta and gamma power changes across brain regions showed bias to specific combinations of stimuli, they did so in distinct manners. While theta power was biased towards the preferred stimulus in the SP task, with almost no region (other than hippocampal areas) showing a higher level during investigation of object stimuli, the gamma power was clearly biased towards the less-preferred stimuli in the EsP and SxP tasks (grouped and male stimuli), while showing a mixed preference between stimuli in the SP task. Thus, as related to gamma power, the social brain may be divided between regions associated with the valence of stimulus (biased towards less-preferred stimuli) and brain regions associated with the type of stimulus. It should be noted that theta rhythmicity is thought to reflect top-down processes, such as arousal and attention, which are regulated by brain wide-active neuromodulators and recruit distributed brain networks[36,56–59]. In contrast, gamma rhythmicity is considered a bottom-up process associated with the synchrony of local inhibitory networks[60–63]. This distinction may explain why theta and gamma rhythms reflect stimulus characteristics in an opposing manner.

It is widely accepted that coherent oscillatory activity of distinct, sometimes remote, brain regions reflects information flow between them[27]. Moreover, coherent oscillations were suggested as a mechanism to bind widespread neuronal ensembles for the purpose of conducting a certain cognitive task[64]. This may be done by providing a temporal window of effective communication (attention) between these ensembles, thus ensuring that a given region provides input when the downstream target is appropriately receptive[43,65]. In accordance with this theory, we

hypothesized that coherent theta and gamma rhythms bind various regions of the social brain in a social context-dependent manner. Thus, different contexts will elicit distinct patterns of coherent oscillatory activity between the various social brain regions, resulting in slightly different types of social information processing and distinct behavioral responses. Our results show that the correlation in coherence change during stimulus investigation was strongest between the two stimuli in each task, even for the EsP and SxP tasks. In contrast, there were weaker correlations, if any, among the three stimuli of the same type (social, grouped, male) or the preferred stimuli (social, isolated, female) across tasks. The fact that the same correlation pattern was observed for the coherence of both theta and gamma rhythms supports the validity and significance of the observation. Moreover, using a Decision trees classifier, we demonstrated that the theta coherence between the recorded areas could generate predictions regarding the social context, but not the specific stimulus, which are accurate above the chance level. The limited accuracy of the model may be attributed to the restricted number of recorded regions. Thus, we expect that a more comprehensive analysis of the coherence within the social brain will be able to generate a much more accurate prediction of the social context. Moreover, GC analysis, representing directional time relationships between various brain regions, also suggests distinct patterns of changes across the various contexts. Altogether, these results are in accordance with the idea that the social brain processes information during stimulus investigation in a context-dependent manner, dictated by the context-dependent pattern of coherence within the network. Such a mechanism may explain how the same stimulus induces distinct patterns of brain activity in different social contexts, which then elicits distinct behavioral responses to a stimulus.

Finally, the coherence changes and GC analyses led us to identify a small subset of brain regions that seem highly influential within the network during the various tasks. Of these, the vDG and AcbC were involved in significant GC changes during all tasks in both the theta and gamma bands, and thus may serve as hubs that influence the activities of other regions. Analysis of LFP power in relation to a battery of specific behavioral events demonstrated that while the small group of brain regions considered showed differential responses as a whole, the vDG was the only region that alone showed statistically significant responses. Together with its strong bias towards specific stimuli, as demonstrated for both theta and gamma power during investigation bouts (Fig. 3h-i), these results suggest a role for the vDG in orchestrating neural activity across the social brain during social behavior. This conclusion agrees with previous studies reporting a central role of the dentate gyrus in social behavior[66–68], and specifically in social discrimination[69,70]. Moreover, the ventral hippocampus was shown to have robust connectivity with various regions of the social brain, including the mPFC, LS, BLA, CeA and nucleus accumbens[71]. Notably, multiple studies have implicated the DG in coding contextual changes. For example, DG

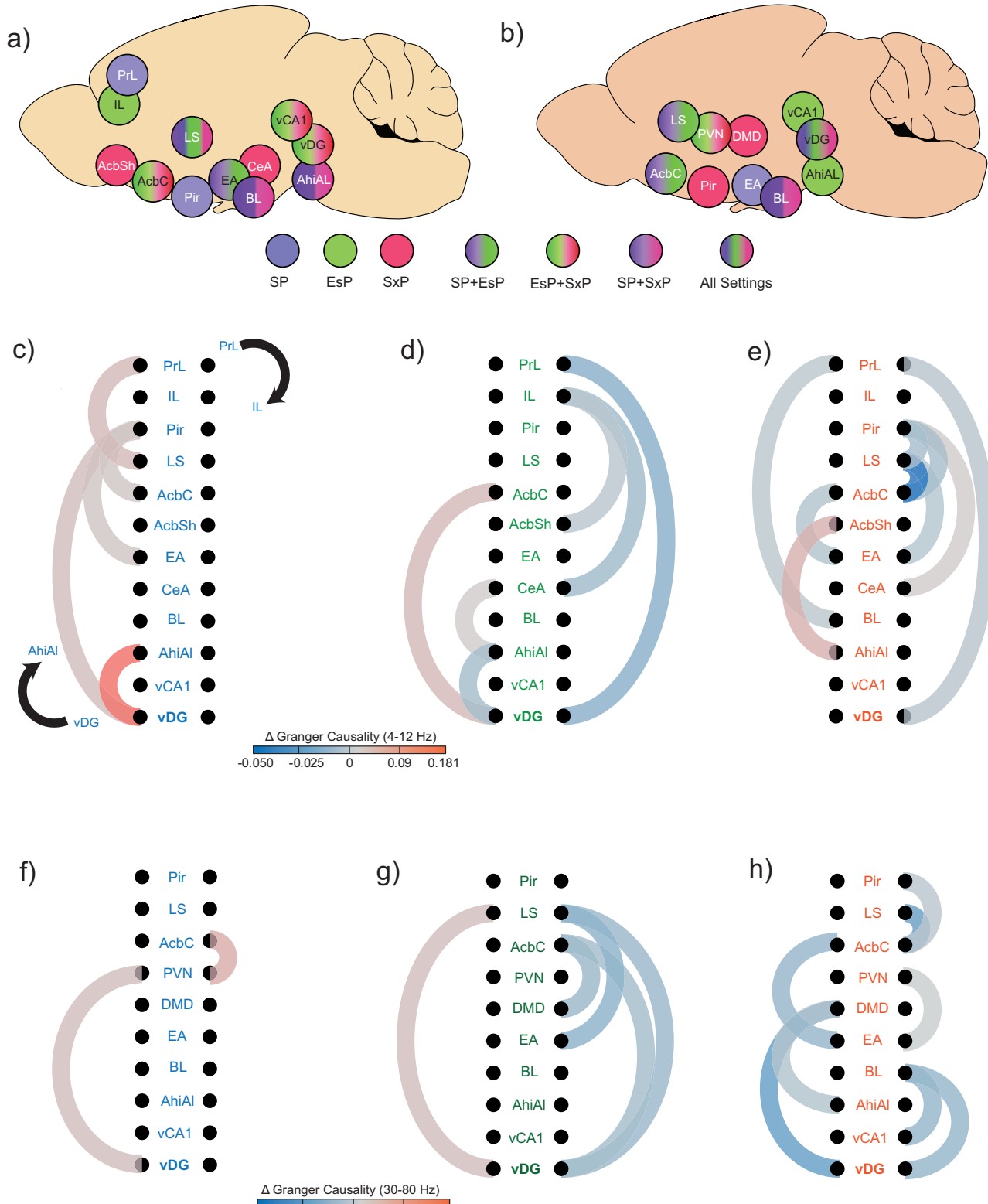

**Fig. 6 Distinct patterns of changes in Granger causality (GC) during the encounter period among tasks. a** Schematic representation of the brain regions which are over-represented among the pairs that exhibited strong (mean ± 1.5*SD) theta coherence bias towards one of the stimuli in any task. The regions are color-coded according to the task in which they were over-represented (see color-code below). **b** As in (**a**), for gamma coherence. **c** Color-coded (see color-code below) schematic representation of significant changes (encounter vs. baseline) in theta band GC during a SP task, among the regions listed in (**a**). The direction of the GC changes is shown by a black arrow (top to bottom on the right and bottom to top on the left). **d** As in (**c**), for EsP. **e** AS in (**c**), for SxP. **f–h** As in (**c–e**), for gamma band. See also Supplementary Fig. 8.

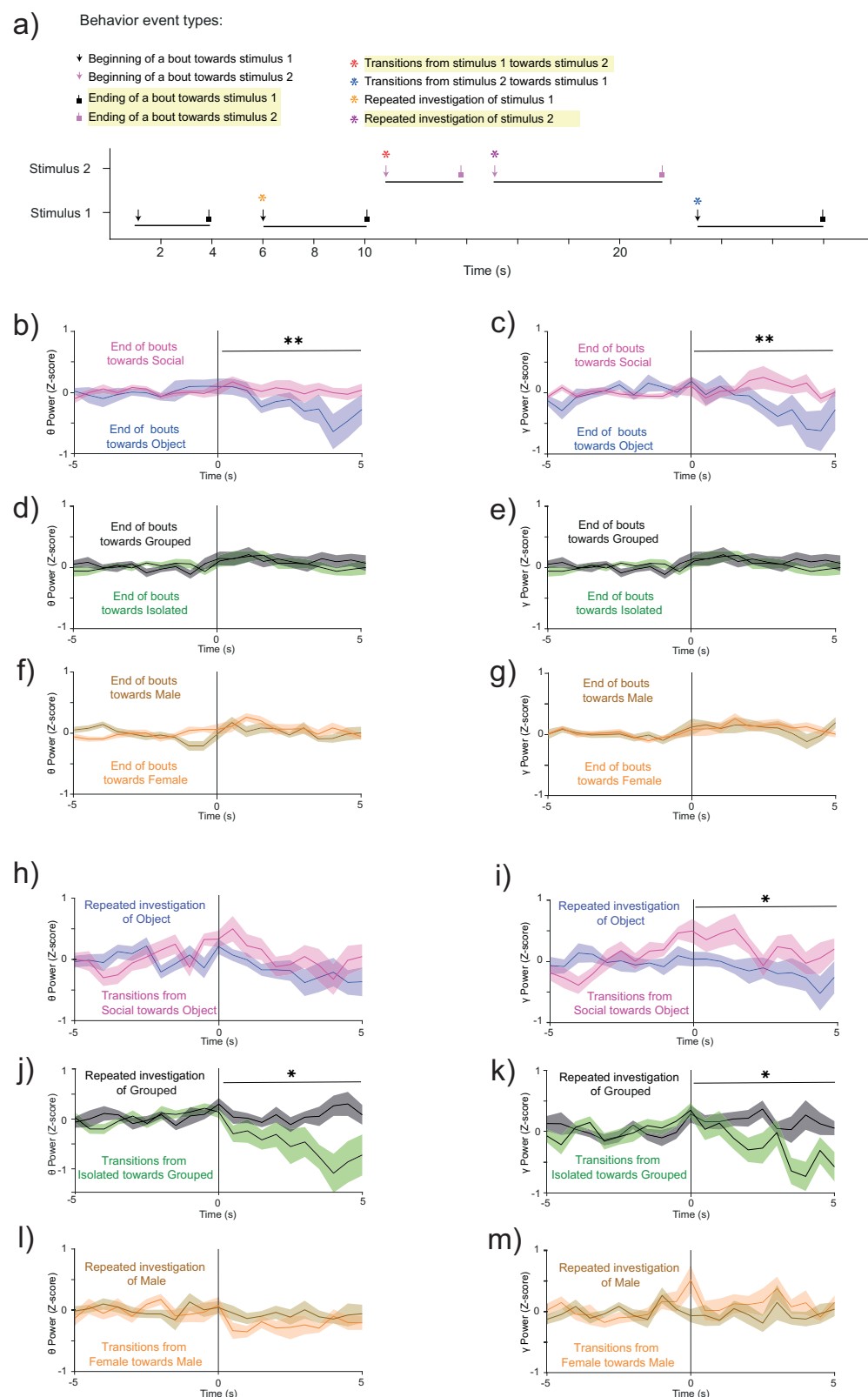

neurons were shown to rapidly detect and encode contextual changes[72], while knocking out NMDA receptors specifically in DG granule cells abolished the ability of mice to distinguish between two similar contexts[48]. Moreover, hypothalamic supra-mammillary neurons projecting to the DG were shown to be activated by contextual novelty[47], while the activity of ventral hippocampal neurons was shown to process information in a

social context-sensitive manner[73]. These studies are thus in line with our findings regarding the involvement of the vDG in context-dependent social behavior.

In conclusion, our results suggest that the distribution of LFP rhythmic activity in the social brain and, most specifically, the synchronization between the various regions is context-specific and may thus mediate context-specific processing of social

**Fig. 7 Context-specific differences in vDG LFP power between specific behavioral events. a** Color-coded scheme of specific behavioral event types. Events showing significant differences in vDG LFP power are highlighted in yellow. **b** Super-imposed traces of the mean (±SEM) Z-score of changes in vDG theta power at the end of investigation bouts towards either the social (pink) or the object stimulus (purple) in the SP task. Time '0' represents the end of the bout, while the 5 s period before time 0 was considered as baseline. **P < 0.01, Student's t test between the mean Z-score values averaged over the 5 s traces, starting at time 0. **c** As in (**b**), for gamma power in the vDG during SP tasks. **P < 0.01, Student's t test. **d–e** As in (**b, c**), for the EsP task. **f–h** as in (**b, c**), for the SxP task. **h-i** As in (**b, c**), for changes in LFP power at the beginning of repeated vs. transitional (between stimuli) investigation bouts of social and object stimuli across SP task sessions. *P < 0.05, Mann–Whitney test. **j–k** As in (**h, i**), for the EsP task. *P < 0.05, Mann–Whitney test. **l, m** As in (**h, i**), for the SxP task. See also Supplementary Fig. 9.

information, leading to social context-dependent social responses and behavior.

## Methods

**Animals.** Adult male and female CD1 mice (12-14 weeks old) were acquired from Envigo (Rehovot, Israel). All mice were housed in groups of 3-5 in a dark/light 12-hour cycle (lights on at 7 pm), with *ad libitum* food and water. Following surgery, implanted mice were housed in isolation so as to not disturb the implanted electrode array (Ear). Experiments were performed in the dark phase of the dark/light cycle in a sound- and electromagnetic noise-attenuated chamber. All experiments were approved by the Institutional Animal Care and Use Committee of the University of Haifa (Ethical approval #616/19).

**Surgery.** Mice were anesthetized using isoflurane (induction 3%, 0.5–0.8% maintenance in 200 mL/min of air; SomnoSuite) and placed over a custom-made heating pad (37 °C) in a standard stereotaxic device (Kopf Instruments, Tujunga, CA). Two burr holes were drilled for placing the ground and reference wires (silver wire, 127 μm, 300-500 Ω; A-M Systems, Carlsborg, WA). Two watch screws (0-80, 1/16", M1.4) were inserted into the temporal bone. The coordinates for Prl (AP = 2 mm, ML = −0.3, DV = −1.8), AcbC (AP = 1, ML = −2.3, DV = −4.7), Pir (AP = −2, ML = −3.3, DV = −5) and CA1 (AP = −3, ML = −3.3, DV = −4.7) were indicated over the left hemisphere using a marker. The skull covering these marked coordinates was removed using a dental drill, and the exposed brain was kept moist with cold, sterile saline. We custom-designed the EAr[49] from 16 individual 50 μm formvar-insulated tungsten wires (50-150 kΩ, #CFW2032882; California Wire Company). Before implantation, the EAr was dipped in DiI (1,1'-Dioctadecyl-3,3,3',3'-tetramethylindocarbocyanine perchlorate; 42364, Sigma-Aldrich) to visualize electrode locations *post-mortem*. The reference and ground wires were inserted into their respective burr holes. The EAr was lowered onto the surface of the exposed brain using a motorized manipulator (MP200; Sutter instruments). The dorsoventral coordinates were marked using the depth of the electrode targeting the PVN (AP = −1 mm, ML = −0.3), which was lowered slowly to −4.7 mm. The EAr and exposed skull with the screws were secured with dental cement (Enamel plus, Micerium). Mice were sub-cutaneously injected with Baytril (5 mg/kg; Bayer) and Norocarp (5 mg/kg; Carprofen, Norbrook Lab) post-surgery and allowed to recover for three days.

**Electrophysiological and video recordings.** Following brief exposure to isoflurane, subjects were attached to the headstage (RHD 32 ch, #C3314, Intan Technologies) through a custom-made Omnetics to Mill-Max adaptor (Mill-Max model 852-10-100-10-001000). Behavior was recorded using a monochromatic camera (30 Hz, Flea3 USB3, Flir) placed above the arena. Electrophysiological recordings were made with the RHD2000 evaluation system using an ultra-thin SPI interface cable connected to the headstage board (RHD 16ch, #C3334, Intan Technologies).

Electrophysiological recordings (sampled at 20 kHz) were synchronized with recorded video using a TTL trigger pulse and by recording camera frame strobes.

**Experiment design.** We recorded the behavior and neural activity of 14 males in the SP task, 13 males in the EsP task, and 11 males in the SxP task (Supplementary Data 2), while targeting 18 distinct brain regions. All the stimuli used for the tasks were unfamiliar to the subject mice. In experiments, the mice were briefly exposed to isoflurane, and the EAr was connected to the evaluation system. After 10 minutes of habituation, the recordings started in the arena (30 × 22 × 35 cm) with empty triangular chambers (12 cm isosceles, 35 cm height), as previously described[52]. The triangular chambers had one face ending with metal mesh (18 mm × 6 cm; 1 cm × 1 cm holes) through which the mice interacted with the stimuli. The test was divided into two 5 min periods, a baseline period (pre-encounter) and a period of encounter with the stimuli. The stimuli for the SP task were a novel group-housed male mouse (social) and a Lego toy (object). For the EsP task, isolated (7–14 days) male and group-housed male mice served as stimuli. Finally, for the SxP task, group-housed male and female mice were used as stimuli. Each subject was evaluated for three sessions of each task. The subjects first performed SP and free interactions, with 10 min between these tasks for three sessions, and then EsP and SxP tasks were performed likewise. Each day four sessions were recorded, two in the morning and two in the afternoon, six hours apart (See Supplementary Fig. 1b). The free interaction data were not used in this study. We excluded sessions from further evaluations when there was a removal of the headstage from the EAr by subjects or in a case of a missing video recording from a session. This accounts for the unequal number of sessions and subjects across tasks.

**Histology.** Subjects were transcardially perfused, and their brains were kept cold in 4% paraformaldehyde for 48 h. Brains were sectioned (50 μm) horizontally (VT 1200 s, Leica). Electrode marks were visualized (DiI coated, Red) against DAPI-stained sections with an epifluorescence microscope (Ti2 eclipse, Nikon). The marks were used to locate the respective brain regions, based on the mouse atlas. Out of all implanted electrodes (256), 9% (23 electrodes from 14 mice) were found to be mistargeted and 36% (93) were non-functional (Supplementary Data 2).

**Behavioral analysis.** Subject behavior was tracked using the TrackRodent algorithm for tethered mice, as previously described[52]. Further parameters of behavior, like duration of interaction, interaction bouts, distance traveled by the subjects, subject speed, transitions between stimuli, and RDI values were calculated as previously described[51,52,74,75].

**Electrophysiological data analysis.** Only brain regions recorded for more than 5 sessions across at least 3 mice were analyzed. All signals were analyzed with codes custom-written in MATLAB 2020a. We excluded the signals recorded during 30 seconds around

stimulus removal and insertion times, to avoid any effect of this action. First, the signals were down-sampled to 5 kHz and low-pass filtered to 300 Hz using a Butterworth filter. The power and time for the different frequencies were estimated using the 'spectrogram' function in MATLAB with the following parameters: Discrete prolate spheroidal sequences (dpss) window = 2 s; overlap = 50%; increments = 0.5 Hz; and time bins = 0.5 s. The power of each frequency band (theta: 4-12 Hz and gamma: 30-80 Hz) was averaged for both the baseline and encounter periods (5-min long each). Changes in theta ($^{\Delta}\theta$P) and gamma ($^{\Delta}\gamma$P) powers for each brain region were defined as the mean difference in power between the encounter and baseline periods. For Z-score analysis of $^{\Delta}\theta$P and $^{\Delta}\gamma$P during investigation bouts for a given stimulus we used the pre-bout 5 s period as baseline, and averaged the Z-score across all bouts with the same stimulus in each session. Notably, throughout the study we have analyzed only investigation bouts that were longer than 2 s, for two reasons: (1) only >2 s bouts showed statistically significant differences between the stimuli in the various tasks (Supplementary Fig. 1f) and (2) only >2 s bouts allow a reliable calculation of theta coherence. LFP power ($^{\Delta}\theta$P and $^{\Delta}\gamma$P) for specific bouts with each stimulus was estimated by calculating the difference between the average power per second during an investigation bout (which was longer than 2 s) during the encounter period and the average power per second for investigation of both empty chambers in the baseline period of the same session, followed by averaging these values over all sessions.

**Coherence analysis**. We used the 'mscohere' function of MATLAB to estimate coherence values using Welch's overlapped averaged periodogram method. The magnitude-squared coherence between two signals, $x$ and $y$, was defined as follows:

$$\text{Coherence}_{xy} = \frac{S_{xy}}{\sqrt{S_{xx} S_{yy}}}$$

where $S_{xy}$ is the cross-power spectral density of $x$ and $y$, $S_{xx}$ is the power spectral density of x and $S_{yy}$ is the power spectral density of y. All coherence analysis was quantified between brain regions pairs involved in at least five sessions of behavior tasks. Coherence for the baseline period was quantified as the average coherence of all brain region pairs for each context (Fig. 4a, c). Changes in coherence ($\Delta\theta$Co and $\Delta\gamma$Co) during the encounter period (Fig. 4b, d) between a pair of brain regions were calculated as follows:

$$\text{Change in Coherence} = \frac{\mu(\text{Coherence}_{\text{encounter}} - \text{Coherence}_{\text{baseline}})}{\sigma(\text{Coherence}_{\text{encounter}} - \text{Coherence}_{\text{baseline}})}$$

where Coherence$_{\text{encounter}}$ is the absolute coherence value between a pair of regions within a frequency band during whole encounter period. Coherence$_{\text{baseline}}$ is the absolute coherence value between a pair of regions within a frequency band during an entire encounter period. The change in coherence for specific bouts ($^{\Delta}\theta$Co and $^{\Delta}\gamma$Co) to each stimulus was estimated by calculating the difference between the average coherence per second during investigation bouts (≥2 s) in the encounter period and the average coherence per second for investigation of both empty chambers during the baseline period of the same session, followed by averaging these values over all sessions.

**Inter-regional pairwise conditional Granger causality**. We employed the multi-variate GC toolbox[55] to calculate GC values separately for baseline and encounter periods between brain regions separately for each task and rhythm. To this end, we selected brain regions most represented among brain region pairs that crossed the mean ± 1.5*SD threshold for the difference in coherence change between preferred and less preferred stimuli in

any task, separately for $^{\Delta}\theta$Co and $^{\Delta}\gamma$Co. For GC analysis, LFP signals were measured at a reduced sampling rate of 500 Hz. We used the "tsdata_to_infocrit" function to determine the model order of the vector autoregressive (VAR) model. The median model order for all three tasks was 38 (Bayesian information criterion). To further fit the VAR model to our multi-session, multivariate LFP data, the "tsdata_to_var" function of LWR (Levinson-Whittle recursion) in the regression mode and a median model order of 38 was used separately for the baseline and encounter periods of each task. Next, we estimated the autocovariance sequence of the fitted VAR model with the "var_to_autocov" function. To maximize the computational efficiency of the function, an acmaxlags of 1500 was chosen. This process did not violate the autocovariance VAR model, as was estimated by the "var_info" function. Finally, we calculated the pairwise conditional frequency-domain multivariate GC matrix using the "autocov_to_spwcgc" function, and summed the GC for the relevant frequency band (theta or gamma) using the "smvgc_to_mvgc" function.

**Neural responses to behavioral events**. We extracted specific behavioral events from the investigation bouts. These include the start and end of an investigation bout, transition from one stimulus to the other or a repeated investigation of the same stimulus (Fig. 7a). We aligned LFP power and behavior events for each stimulus by calculating the mean power across five seconds before and five seconds after the beginning (or end) of all investigation bouts in a session, using 0.5-s bins. Furthermore, for each event, the mean power was normalized using Z-score analysis, where the pre-bout 5-s period served as baseline (Supplementary Data 3).

**Statistics and reproducibility**. Statistical analysis was performed using GraphPad Prism 9.5. To test for the normal distribution of the data, we used the Kolmogorov-Smirnov and Shapiro-Wilk tests. Supplementary Data 4 summarizes the specific tests conducted for each figure. A paired t-test or Wilcoxon matched-pairs signed rank test was used to compare different stimuli or conditions for the same group. An unpaired t-test or Mann-Whitney test was performed to compare a parameter between distinct groups. For comparison among multiple groups and parameters, ANOVA (normal distribution), Welch's ANOVA (assuming unequal variance), and Kruskal-Wallis test (non-normal distribution) were applied to the data. If a main effect or interaction were found in the tests above, Šídák's test, Dunnett's T3 test or Dunn's *post-hoc* multiple comparison corrections were applied. Repeated measures ANOVA or a Friedman test was used to compare multiple groups and parameters with repeated variables. When main effects were observed in the above tests, Šídák's or Dunn's tests were used for multiple comparisons corrections, respectively. Additionally, for comparison of two factors and the interaction between them, from multiple groups and parameters where one of the factors has repeated measurements, was performed using two-way ANOVA (no missing variables) or mixed-models ANOVA (Restricted maximum likelihood model, REML). The ANOVA tests were followed by Šídák's multiple comparison test if main effects or interactions were found. The association between two groups or parameters was compared with either Pearson's or Spearman's tests. A binomial distribution test was performed to compare the probability of brain regions representing above-chance levels for specific parameters. For each analysis, sample size is denoted below the corresponding figure.

**Decision tree classifier model**
*Data normalization—subtracting the mean value for each brain regions pair per mouse*. The data from two mice (total 14) were ignored as they had less than 40 recorded brain regions pairs

(out of 99). The mean value of each pair was computed and subtracted for each mouse separately. This helped to reduce the variability of the measurements across mice and improved classification accuracy. To reduce over-representation of a single stimulus in the computation of the mean value for a pair, we first averaged the mean value per stimulus (for a specific mouse) and then subtracted the average of these means.

*Averaging bouts.* The average bout for each stimulus was computed for each session.

*Data Imputation.* For each mouse, a slightly different set of brain areas were recorded due to slight inaccuracies in placing the electrodes and slight difference in the individual mouse anatomy. This resulted in missing entries from some of the brain regions pairs. We used a data imputation strategy to restore these missing entries. Note that before this step, we subtracted the mean value per brain region per mouse and averaged all the bouts from the same stimuli of the same session. The imputation algorithm is based on the MICE algorithm[76]. It is an iterative algorithm. In each iteration, it estimates the missing entries by a linear combination of (some of) the other entries. The used data imputation algorithm was defined as follows:

1. For each missing value of brain pair i in bout b, $(bp_{i,b})$, replace $bp_{i,b}$ with the average value of $bp_i$ across the valid values $bp_i$ (from all bouts of all mice with a valid measurement of $bp_i$).
2. For each $bp_i$ (order of brain pair is randomized):
   a. Randomly choose a set of predicting brain pairs $\{bp_j\}$ such that $bp_i \notin \{bp_j\}$ and $|\{bp_j\}| < 0.5*$number_of_equations. Where the number_of_equations equals to the number of (averaged) bouts from all the mice (66 predicting brain pairs as the number of bouts in our dataset is 131 average bouts).
   b. Compute linear regression coefficients {aj} (by least square method) to minimize: $\underset{\{a_j\}}{\text{argmin}} \sum_b (bp_{i,b} - (a_0 + \sum_j a_j bp_{j,b}))^2$
   c. For each bout b in which $bp_{i,b}$ was not measured in-vivo, replace it with its estimation: $a_0 + \sum_j a_j bp_{j,b}$
3. Repeat step 2 for 20 iterations.

Code was implemented in MATLAB 2021a.

*Classification and computation of confusion matrixes.* We used MATLAB 's *TreeBagger()* function to train a multi-class Random forest classifier for discriminating between a pair of stimuli in the SP test (social vs. object), three contexts (SP. EsP and SxP) or between six stimuli over all three contexts. We used 80 random trees (a parameter of the TreeBagger function). We used cross-validation with "one mouse leave out" strategy to compute a confusion matrix for each mouse based on a training set that includes examples from all of the other mice. We balanced the training set to have the same number of examples from each class by randomly removing some of the training examples. Since both the balancing and the Random forest algorithm have a random component, we repeated the estimation of confusion matrixes 100 times (for each mouse) to better estimate the confusion matrixes. We then summed up all of the confusion matrixes (totally 1200 confusion matrixes: 12 mice and 100 confusion matrixes per mouse) and computed for each pair of classes (i,j) the percent of cases where the prediction was i for bouts of class j.

*Statistical analysis of the models.* All tests were corrected for multiple comparisons using FDR corrections[77]. To compute

p-values, we used the average (over 100 iterations) confusion matrix for each mouse (totally 12 confusion matrixes in which each cell *i,j* represents the % of predictions of class i for bouts of class j) and compare those with a set of random confusion matrixes which were generated by the same procedure except for replacing the trained classifier with a random classifier. This random classifier generated random labels with uniform distribution. To better describe the distribution of the random confusion matrixes, we generated 83 random confusion matrixes per mouse (each one of them is an average of 100 confusion matrixes). Then, the p-value for each cell in the confusion matrix was computed separately by comparing the 12 values from the confusion matrix of the trained classifier to the $83 \times 12 = 996$ values from the confusion matrixes of the random classifier using Mann–Whitney U test. In case a mouse did not have a bout from a specific class, this mouse was ignored in the computation of the p-value for the cells of this ground truth class.

All details of the statistical analyses appear in Supplementary Data 4, arranged according to the various figures.

**Reporting summary**. Further information on research design is available in the Nature Portfolio Reporting Summary linked to this article.

## Data availability

All source data underlying the graphs presented in the figures have been provided as Supplementary Data 1. For Fig. 5d, e and Supplementary Fig. 7a–f (decision tree model data) please use the MATLAB codes and data provided in *Zenodo* https://doi.org/10.5281/zenodo.10232644[78]. All raw data generated from the above experiments is deposited in *Zenodo* (https://zenodo.org/), https://doi.org/10.5281/zenodo.10232693[79].

## Code availability

All Custom codes written in MATLAB used for analyzing the generated data is deposited in *Zenodo* (https://zenodo.org/), https://doi.org/10.5281/zenodo.10232644[79].

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

## Acknowledgements

The authors thank Boris Shklyar, Head of the Bio-imaging Unit and Dr. Maya Lalzar, Head of the Bioinformatics Unit of the Faculty of Natural Sciences of the University of Haifa for their guidance and technical assistance. We also thank Eng. Alex Bizer, the Experimental Systems Engineer of the Faculty of Natural Sciences of the University of Haifa, for his help. This study was supported by the ISF-NSFC joint research program (grant No. 3459/20), the Israel Science Foundation (Grants No. 1361/17 and 2220/22), the Ministry of Science, Technology and Space of Israel (Grant No. 3-12068), the Ministry of Health of Israel (grant #3-18380), the German Research Foundation (DFG) (GR 3619/16-1 and SH 752/2-1), the Congressionally Directed Medical Research Programs (CDMRP) (grant No. AR210005) and the United States-Israel Binational Science Foundation (grant No. 2019186).

## Author contributions

A.N.M.: Formal analysis, Investigation, Methodology, Validation, Visualization, Writing—original draft, and Writing—review & editing; D.P.: Formal analysis, and software.; S.N.: Data curation, Project administration, Software, Validation, Visualization, Writing—original draft, and Writing—review & editing. S.W.: Conceptualization, Funding acquisition, Project administration, Resources, Supervision, Writing—original draft, and Writing—review & editing

## Competing interests

The authors declare no competing interests.
