## [Peer Review File · Communications Biology]

Reviewers' comments:

Reviewer #1 (Remarks to the Author):

Mohapatra et al. examined how LFP patterns from multiple brain regions are related to social and emotional experiences. Overall, the study is well designed, and their datasets and analyses are rigid and highly reliable. I have several suggestions to improve the manuscript.

Major comment:

- (1) A major finding of this study is the importance of the vDG and the authors mentioned possible roles of the vDG (Line 542-552). In addition, could the authors discuss their results based on any anatomical connections? For example, we easily imagine that the downstream regions of vDG are vCA1 and vCA3 but their results did not detect prominent functional connections. On the other hand, Figure 6C shows strong information transfer from vDG to AhiAI. Are there any direct connections between these regions? If possible, please discuss them.
- (2) Please describe the numbers of samples in all graphs (not only the degree of freedom). At Line 619, they describe that some electrodes are missed from targeted regions. In that case, how did they apply the Random forest model to the missing data?
- (3) Are there any relationships between electrode-electrode distance and coherence?

Minor comment:

- (1) Please specify the sex of mice in the Abstract.
- (2) Please consider to make a conclusion sentence at the end of the Abstract as they just described results (Line 12).
- (3) Figure 2A: I do not clearly find the difference in the spectrums between baseline and encounter. It may be useful to use different color scales?
- (4) Figure 3H-I: 3D plots are beautiful but it is hard to understand whether the vDG is plotted. Please consider to add scalars or 2D-projected plots.
- (5) Line 249-250: Fig. 4B and Fig. 4C should be Fig. 3H and Fig. 3I?
- (6) Line 388: Fig. 6-H should be Fig. 6F-H?
- (7) Line 448: Fig. 3B-C. it is correct?
- (8) Figure S5: "Cortex" should be "Neocortex" as this term includes hippocampus as well.
- (9) Figure S5: "Hippocampus" should be "Hippocampal formation" as this term does not include vDG in a strict sense.

Reviewer #2 (Remarks to the Author):

Fig. 3A-F

a) 3A-C: The changes in theta power show large trial-by-trial variance between the conditions. Can it be excluded that the observed changes are random and unrelated to the behavior?

b) Page 10: Line 209 "This analysis thus suggests a bias in the response towards specific stimuli, in a task-specific manner."

It is not clear how the authors tested for the significant difference between the measurements in 3A-F to reach this conclusion.

Fig. 3G

It is not clear what each dot represents. Please add to the legend.

Fig. 3H-I

It is visually hard to distinguish the 3 axes and the arrangement of the points in space, which occludes an evaluation of the special location of vDG on the plot. Is there a way to rotate the plot and to find a better angle to highlight vDGs role?

Fig. 4H-I

The scatter plots for the correlation analysis should be provided.

Fig. S4

The analysis similar to S4G seems to be missing for S4E and S4F.

Fig. 7

Figure 7 is unclear. Why was a 3 s analysis time window at the end of the bout chosen? The current selection seems arbitrary in relation to the behavior. E.g., Fig. 7F and Fig. 7I exhibit potentially significantly different changes in power between the behavioral event types but are currently not considered as relevant. The time window selection needs to be clarified.

Internal state

It is not entirely clear what the authors mean by internal state. A definition would be useful.

Behavior

The behavioral classification and onsets time definition for behavioral bouts (e.g. how were interactions scored etc.) is currently unclear and needs to be described in more detail.

Figure references

line 248-252 (wrong figure reference), line 259 (wrong figure reference), line 388 (wrong figure reference)

Reviewer #3 (Remarks to the Author):

Using LFP recordings targeted to multiple brain regions involved in social behaviour, the authors of this interesting study revealed a complex organization of inter-regional synchronization at theta and gamma frequency bands during distinct social behavioural paradigms. A strength of the study is the large number of multi-site recordings and the wealth of data analyses applied to the recorded data. As a limitation of the study I would like to point out that it was challenging to extract a consistent narrative from the paper in its current form. The presented results are without doubt interesting, but the complex nature of the results makes it hard to follow the authors' line of thought throughout. What precisely is the role of theta/gamma in linking the various regions in a context/stimulus-dependent manner? Granted that the authors cannot conclusively answer that question with the data at hand, a clearer interpretation of the results would certainly help the reader. In addition to this rather general comment I have some specific recommendations that might further improve the manuscript:

Major points:

1. Abstract: While the authors list their key findings it is somewhat hard to get the main message of the paper from the abstract in its current form. Specifically, in line 8, they write that 'rhythmicity across all tasks was dominated by a general internal state'. I find the expression 'general internal state' to be somewhat vague. It would be better if the authors explicitly described the state. Moreover, the abstract would benefit from a concluding statement that explains the meaning of the observed phenomena, even if that interpretation might be to some extent speculative.
2. Coherence analysis during behavioural bouts: Did the authors consider that different epoch lengths used for the coherence estimation might have affected the results? This could be countered by repeating the coherence analysis after first restricting the analyzed period to the length of the recording session with the shortest total interaction time. Moreover, volume conduction might differently affect coherence between any two given recording sites, depending on their anatomical proximity. The authors should run additional controls for that, for example by computing directed coherence or by using another method that reduces the potential impact of volume conduction.
3. Decoding model: The authors seem to have implemented a decoding approach in which coherence values are normalized averaged for each mouse over all behavioural bouts. Does successful classification depend on the chosen model? The analysis could be repeated with different decoding models to confirm the result. Moreover, would the result hold when the analysis was done for each mouse independently? This could be tested, for instance, by training/predicting with individual bouts separately for each mouse.
4. Granger causality analysis: It would be informative to additionally compute GC specifically for the bouts instead of taking the entire task epoch.

Minor points:

5. The behavioural results are based on male mice only. The authors honestly report that some of the tasks did not work with females, which is very much appreciated. A brief discussion on the potential reasons for the sex-dependence of the behavioural performance would be beneficial.
6. Line 249: This should probably read "Fig. 3H"? Same for Fig. 3I in line 250.
7. Line 347: 'First, we validated that the model achieved good (~60%) and significant accuracy in

predicting the social stimulus vs. object in the SP task using either $\Delta \theta_{Co}$ or $\Delta \gamma_{Co}$. Notably, the classification of the object vs. social was not accurate, suggesting that the presence of the social stimulus (social context) mask the object classification (Fig. S7A-B)'. I'm confused by this sentence. On the one hand, the authors state that the decoder achieved 'good (~60%) and significant accuracy', on the other hand the 'classification of object vs. social was not accurate'. Am I missing something or do both statements contradict each other? It would be advisable to make this statement clearer.

8. Line 751: 'or between stimuli (6 classes) or between.' This sentence is incomplete.

9. Figure 1: No statistical details are given for the comparisons in M-O. It is quite obvious from the plots that no difference was found, and I presume the same Kruskal-Wallis procedure as in Fig. 1P was applied? The results of these tests should be given in the figure legend.

10. Figure 2:

- The heatmaps in panel A are hard to interpret at the current scaling used. It might be beneficial to narrow the visualized range, or given the different absolute power of theta and gamma, to show the theta and gamma ranges at separate scaling (as insets, perhaps).
- panels C-F show the tasks in the previously established order (SP, EsP, SxP) while the order is reversed in G and H. It would be more consistent to plot G and H in the same order.

11. Figure 3: The 3D plots in H and I are hard to interpret as it is difficult to properly localize the individual data points in the 3D space. This, of course, is an inherent problem with 3D plots, and very likely the authors already explored alternative visualization methods. Still, it might be worth considering a different way of showing the data, perhaps by plotting in 2D and showing the 3rd dimension on a continuous colour scale? The point is that the authors' central argument of different clustering for theta and gamma in this state space is difficult to confirm visually in the current plots.

Response to Reviewers notes

We thank the Reviewers for their excellent and helpful comments and suggestions. We believe that thanks to them the revised manuscript, in which we did our best to address all the reviewer's notes, is significantly improved.

Below we provide a point-by-point response to the concerns that were raised.

Reviewer #1 (Remarks to the Author):

Mohapatra et al. examined how LFP patterns from multiple brain regions are related to social and emotional experiences. Overall, the study is well designed, and their datasets and analyses are rigid and highly reliable. I have several suggestions to improve the manuscript.

Major comment:

(1) A major finding of this study is the importance of the vDG and the authors mentioned possible roles of the vDG (Line 542-552). In addition, could the authors discuss their results based on any anatomical connections? For example, we easily imagine that the downstream regions of vDG are vCA1 and vCA3 but their results did not detect prominent functional connections. On the other hand, Figure 6C shows strong information transfer from vDG to AhiAl. Are there any direct connections between these regions? If possible, please discuss them.

Response: Since there are very few studies on vDG connectivity, we prefer not to speculate on this subject. We did, however, cited in the Discussion section a recent paper which explored the connectivity of the ventral CA1 and showed that it is strongly connected to many of the regions explored by us, as followed:

“Moreover, the ventral hippocampus was shown to have robust connectivity with various regions of the social brain, including the mPFC, LS, BLA, CeA and nucleus accumbens [73].”

(2) Please describe the numbers of samples in all graphs (not only the degree of freedom).

Response: We have now included the number of sessions, brain pairs or brain regions as appropriate in each item of the figures.

At Line 619, they describe that some electrodes are missed from targeted regions. In that case, how did they apply the Random forest model to the missing data?

Response: The reviewer is right, and some electrodes are missed from targeted regions, which we highlight in the Histology section in Methods. We explained there the issue of unequal data, and how we used an imputation algorithm to fill missing values after data normalization step. These steps were detailed in Decision Tree classifier model section of the Methods.

(3) Are there any relationships between electrode-electrode distance and coherence?

Response: We have now added a Supplementary Fig. 4D-G (shown below) where we compare, for SP task, the correlation of coherence values to increasing distance between each pair of brain regions. Theta and gamma coherence during baseline (A and B) slightly correlated to increasing distance between pairs, with relative reduction in coherence values. This correlation during resting is expected, due to volume conduction. However, there was no effect of distance on both theta or gamma coherence change (C and D) during the encounter period. Therefore, we conclude that, unlike resting state, during social encounter the effect of volume conduction on synchronous rhythmicity is negligible and the coherence is affected mainly by the social behavior and context.

Minor comment:

(1) Please specify the sex of mice in the Abstract.

Response: Done

(2) Please consider to make a conclusion sentence at the end of the Abstract as they just described results (Line 12).

Response: Done

(3) Figure 2A: I do not clearly find the difference in the spectrums between baseline and encounter. It may be useful to use different color scales?

Response: We have Corrected the scale of spectrogram plot for Fig. 2A.

(4) Figure 3H-I: 3D plots are beautiful but it is hard to understand whether the vDG is plotted.

Please consider to add scalars or 2D-projected plots.

Response: We have added a supplementary figure showing the requested 2D plots (Supplementary Fig. 3G-H), as shown below, to the revised manuscript. The labels of each brain region are similar to Fig.3H, I to provide consistent and clear illustration to description in the text.

(5) Line 249-250: Fig. 4B and Fig. 4C should be Fig. 3H and Fig. 3I?

Response: Corrected

(6) Line 388: Fig. 6-H should be Fig. 6F-H?

Response: Corrected

(7) Line 448: Fig. 3B-C. it is correct?

Response: The reviewer pointed this correctly in line 450, it should be Fig 3 H-I rather than 3B-C. This was corrected.

(8) Figure S5: “Cortex” should be “Neocortex” as this term includes hippocampus as well.

Response: Corrected

(9) Figure S5: “Hippocampus” should be “Hippocampal formation” as this term does not include vDG in a strict sense.

Response: Corrected

Reviewer #2 (Remarks to the Author):

Fig. 3A-F

a) 3A-C: The changes in theta power show large trial-by-trial variance between the conditions. Can it be excluded that the observed changes are random and unrelated to the behavior?

Response: In order to validate whether variability of theta power between trials was statistically significant across animals, we plotted the average theta power (all areas pooled) for each session across all subjects during encounter period of SP task. There is observable difference for few individual subjects in subsequent sessions. However, there was no statistical difference between the sessions (trials) (G: Paired t test, 14 subjects, $t(13) = 1.722$, $P = 1.087$). Further the variance in change in Theta power of each brain region within a session was not different in subsequent session (H: Wilcoxon test, 14 subjects, $W = -29$, $P = 0.391$). Similar results were obtained for gamma power. These results, which are shown before, are displayed in Fig 2G-H of the revised manuscript.

b) Page 10: Line 209 “This analysis thus suggests a bias in the response towards specific stimuli, in a task-specific manner.”

It is not clear how the authors tested for the significant difference between the measurements in 3A-F to reach this conclusion.

Response: We have changed the example shown in Fig. 3A-F for a better one (see below), and quantitatively analyzed the bias using Z-score analysis, as shown in Supplementary Fig. 4A-C (also shown below).

Fig. 3G

It is not clear what each dot represents. Please add to the legend.

Response: We have added the following statement to the legend of the Figure, to clarify the correlation plots. “Each circle represents $\Delta\theta P$ and corresponding RDI during a single session.”

Fig. 3H-I

It is visually hard to distinguish the 3 axes and the arrangement of the points in space, which occludes an evaluation of the special location of vDG on the plot. Is there a way to rotate the plot and to find a better angle to highlight vDGs role?

Response: We have added a supplementary figure showing the requested 2D plots (Supplementary Fig. 3G-H) to the revised manuscript, as shown above in the response to reviewer #1. The labels of each brain region are similar to Fig.3H, I to provide consistent and clear illustration to description in the text.

Fig. 4H-I

The scatter plots for the correlation analysis should be provided.

Response: We have now provided the requested scatter plots (shown below) in Supplementary Fig 6 of the revised manuscript.

Fig. S4

The analysis similar to S4G seems to be missing for S4E and S4F.

Response: We deleted Fig. S4G, it is not necessary anymore.

Fig. 7

Figure 7 is unclear. Why was a 3 s analysis time window at the end of the bout chosen? The current selection seems arbitrary in relation to the behavior. E.g., Fig. 7F and Fig. 7I exhibit potentially significantly different changes in power between the behavioral event types but are currently not considered as relevant. The time window selection needs to be clarified.

Response: We thank the reviewer for this helpful note. We have reanalyzed the data corresponding to this figure, where we consider the whole 5-s long events between the stimuli of each task. As shown below, the significance of the results didn't change, although Fig. 7I came significant too. The figure and the corresponding text and legend have been updated with these results.

Internal state

It is not entirely clear what the authors mean by internal state. A definition would be useful.

Response: We have now defined the term internal state in the 1st paragraph of the Introduction section: “Such context dependent hidden processes which determine how brains respond to inputs and produce behavioral outputs are defined as internal states and include arousal, motivation, emotion and varying homeostatic needs”

Behavior

The behavioral classification and onsets time definition for behavioral bouts (e.g. how were interactions scored etc.) is currently unclear and needs to be described in more detail.

Response: We have done that in the Methods section, as detailed below:

“Neural responses to behavioral events

We extracted specific behavioral events from the investigation bouts. These include the start and end of an investigation bout, transition from one stimulus to the other or a repeated investigation of the same stimulus (Fig. 7A). We aligned LFP power and behavior events for each stimulus by calculating the mean power across five seconds before and five seconds after the beginning (or end) of all investigation bouts in a session, using 0.5-s bins. Furthermore, for each event, the mean power was normalized using Z-score analysis, where the pre-bout 5-s period served as baseline (Supplementary Table 2). “

Figure references

line 248-252 (wrong figure reference), line 259 (wrong figure reference), line 388 (wrong figure reference)

Response: All these mistakes are now corrected.

Reviewer #3 (Remarks to the Author):

Using LFP recordings targeted to multiple brain regions involved in social behaviour, the authors of this interesting study revealed a complex organization of inter-regional synchronization at theta and gamma frequency bands during distinct social behavioural paradigms. A strength of the study is the large number of multi-site recordings and the wealth of data analyses applied to the recorded data. As a limitation of the study I would like to point out that it was challenging to extract a consistent narrative from the paper in its current form. The presented results are without doubt interesting, but the complex nature of the results makes it hard to follow the

authors' line of thought throughout. What precisely is the role of theta/gamma in linking the various regions in a context/stimulus-dependent manner?

Granted that the authors cannot conclusively answer that question with the data at hand, a clearer interpretation of the results would certainly help the reader.

Response: We have added the following paragraph to the Discussion section of the revised manuscript:

“It is widely accepted that coherent oscillatory activity of distinct, sometimes remote, brain regions reflects information flow between them [65]. Moreover, coherent oscillations were suggested as a mechanism to bind widespread neuronal ensembles for the purpose of conducting a certain cognitive task [66]. This may be done by providing a temporal window of effective communication (attention) between these ensembles, thus ensuring that that a given region provides input when the downstream target is appropriately receptive [44, 67]. In accordance with this theory, we hypothesized that coherent theta and gamma rhythms bind various regions of the social brain in a social context-dependent manner. Thus, different contexts will elicit distinct patterns of coherent oscillatory activity between the various social brain regions, resulting in slightly different types of social information processing and distinct behavioral responses”

In addition to this rather general comment I have some specific recommendations that might further improve the manuscript:

Major points:

1. Abstract: While the authors list their key findings it is somewhat hard to get the main message of the paper from the abstract in its current form. Specifically, in line 8, they write that ‘rhythmicity across all tasks was dominated by a general internal state’. I find the expression ‘general internal state’ to be somewhat vague. It would be better if the authors explicitly described the state.

Response: We have rephrased the abstract and wrote “dominated by a global internal state”. We have also added to the Introduction a sentence stating that “Moreover, high theta and gamma rhythmicity was associated with various internal states, such as avoidance, fear, anxiety and attention”. We have also added a definition of internal state and the possible states

involved in the 1st paragraph of the Introduction section. However, we cannot identify the specific internal state, whether it is arousal, attention or social affiliation, with the data that we currently have.

Moreover, the abstract would benefit from a concluding statement that explains the meaning of the observed phenomena, even if that interpretation might be to some extent speculative.

Response: We thank the reviewer for this helpful suggestion. We have now added a concluding sentence to the abstract of the revised manuscript.

2. Coherence analysis during behavioural bouts: Did the authors consider that different epoch lengths used for the coherence estimation might have affected the results? This could be countered by repeating the coherence analysis after first restricting the analyzed period to the length of the recording session with the shortest total interaction time.

Response: We have done the coherence analysis in epochs of 2 seconds for several reason. First, this is the minimal duration of an investigation bout that we have analyzed throughout the paper, because only <2s bouts showed a significant difference between the stimuli (see Supp. Fig. 1F). Thus, we have actually adopted the reviewer's suggestion to use the shortest duration of investigation bout. Second, this is the minimal time that allows reliable calculation of theta coherence, according to Gourevitch et al., <https://doi.org/10.1152/jn.01075.2009>.

Moreover, volume conduction might differently affect coherence between any two given recording sites, depending on their anatomical proximity. The authors should run additional controls for that, for example by computing directed coherence or by using another method that reduces the potential impact of volume conduction.

Response: We thank the reviewer for this helpful comment. We have calculated the correlation between the distance amongst the regions and the coherence or change in coherence between them. While theta and gamma coherence at baseline showed weak but significant (or borderline significant) correlation with the distance between the brain regions in each couple, no such correlation was observed for the change in theta or gamma coherence ($\Delta\theta\text{Co}$, $\Delta\gamma\text{Co}$) during the encounter. These results, shown in Supplementary Fig. 4D-G of the revised manuscript, suggest no effect of volume conductance on the coherence change during the encounter.

3. Decoding model: The authors seem to have implemented a decoding approach in which coherence values are normalized averaged for each mouse over all behavioural bouts. Does successful classification depend on the chosen model? The analysis could be repeated with different decoding models to confirm the result. Moreover, would the result hold when the analysis was done for each mouse independently? This could be tested, for instance, by training/predicting with individual bouts separately for each mouse.

Response: Our classification model included a step of averaging the coherence data of bouts from each stimulus in a single session, resulting in two average bouts for each session. These average bouts were then used to train a classifier. The purpose of this averaging was to reduce the noise that might exist in the data of a single bout. An additional advantage of this averaging was a reduction of bias in the classifier results as we now explain. We say that a classifier is biased if the probability of predicting class A is generally higher than predicting class B. This might happen if the training set includes more examples of class A than examples of class B. To reduce the bias of the classifier we included a balancing step that made sure that the training set included an equal number of training examples from each class. This was achieved by randomly removing over-expressed classes from the training set. While making the training set data balanced, this strategy also decreased the size of the training set.

Another potential source of classification bias might occur in case the mouse identity is correlated with a specific class. For example, say bouts from mouse1 and mouse2 are included in the training set. If mouse1 has many examples of class A but not from class B, while mouse2 has many examples from class B but not from A, then the classifier might detect the mouse identity from the bouts data and rely on the mouse identity to determine if the bout is from class A or class B. The averaging strategy reduces the risk of such bias significantly since each session will generate 2 average bouts, no matter how many original bouts were in that session.

To test the ability to classify non-averaged bouts instead of average bouts we did two experiments:

- 1) We took the non-averaged bouts and used the balancing method described above to make sure that each class was represented with the same number of examples in the training set. This resulted in a comparable accuracy for the Social vs Object classification task (Fig 1). However, it resulted in low accuracy for the 3 classes context class and what seems as a bias in the output classifier (Fig 2). We believe that this bias was caused by a correlation between

mouse identity and the output class as explained in paragraph 6 above. To further test this we tried another balancing approach which shall be now described.

- 2) To reduce the correlation between mouse identity and the target class, we tested a different balancing method and took the same number of (non-averaged) bouts from each class of each mouse i.e. bouts from class k of mouse i have the same number of occurrences in the training set as class c of mouse j for all k,i,c,j . While this type of balancing removed the correlation between the mouse identity and the class of the bout, it resulted in a much smaller training set since data of many bouts was randomly removed from the training set to allow mice with a small number of bouts for a single class to participate in the training set. This experiment resulted in most of the bias being fixed but the classifier accuracy remains low (Fig 3).

To conclude these experiments, the low accuracy we achieved for the context classification of non-averaged bouts may be due to higher noise in these non-averaged bouts (in comparison to the averaged bouts) or due to a small training set or due to a combination of both. In future work, additional machine learning approaches might be tested to cope with this challenge.

Another point raised by the reviewer is the option of training a classifier on data from a single mouse. The challenge with such a task is that the training for a single mouse would be very small. Using machine learning methods that are designed to cope with very small databases might be a subject of future work.

Figure 1: Classifying social vs object using non-averaged bouts data, Left: using $\Delta\theta Co$. Right: using $\Delta\gamma Co$.

Figure 2: Classifying context using non-averaged bouts data while balancing only for the number of bouts from each class in the training set, Left: using $\Delta\theta Co$. Right: using $\Delta\gamma Co$.

Figure 3: Classifying non-averaged bouts, Balancing both mouse identity and class. Left: using $\Delta\theta Co$. Right: using $\Delta\gamma Co$.

4. Granger causality analysis: It would be informative to additionally compute GC specifically for the bouts instead of taking the entire task epoch.

Response: Unfortunately, there are not enough bouts in a typical session in order to do the analysis as suggested by the reviewer.

Minor points:

5. The behavioural results are based on male mice only. The authors honestly report that some of the tasks did not work with females, which is very much appreciated. A brief discussion on the potential reasons for the sex-dependence of the behavioural performance would be beneficial.

Response: Since this is not one of the main themes of this work, we believe that getting into the sex-dependent behavior would be distracting and would not help the reader. We are working on a paper that specifically describes these sex-dependent differences in details and we believe that this would be a better place for such discussion.

6. Line 249: This should probably read “Fig. 3H”? Same for Fig. 3I in line 250.

Response: This mistake was corrected.

7. Line 347: ‘First, we validated that the model achieved good (~60%) and significant accuracy in predicting the social stimulus vs. object in the SP task using either $\Delta \theta_{Co}$ or $\Delta \gamma_{Co}$. Notably, the classification of the object vs. social was not accurate, suggesting that the presence of the social stimulus (social context) mask the object classification (Fig. S7A-B)’. I’m confused by this sentence. On the one hand, the authors state that the decoder achieved ‘good (~60%) and significant accuracy’, on the other hand the ‘classification of object vs. social was not accurate’. Am I missing something or do both statements contradict each other? It would be advisable to make this statement clearer.

Response: We thank the reviewer for this helpful comment. We have rephrased this paragraph to avoid confusion.

8. Line 751: ‘or between stimuli (6 classes) or between.’ This sentence is incomplete.

Response: This was corrected.

9. Figure 1: No statistical details are given for the comparisons in M-O. It is quite obvious from the plots that no difference was found, and I presume the same Kruskal-Wallis procedure as in Fig. 1P was applied? The results of these tests should be given in the figure legend.

Response: We have added the statistical details to the legend, as suggested by the reviewer.

10. Figure 2:

- The heatmaps in panel A are hard to interpret at the current scaling used. It might be beneficial to narrow the visualized range, or given the different absolute power of theta and gamma, to show the theta and gamma ranges at separate scaling (as insets, perhaps).

Response: We have updated the colors for this, by reducing the scale of the heatmap as also suggested by reviewer #1.

- panels C-F show the tasks in the previously established order (SP, EsP, SxP) while the order is reversed in G and H. It would be more consistent to plot G and H in the same order.

Response: This was corrected as suggested by the reviewer.

11. Figure 3: The 3D plots in H and I are hard to interpret as it is difficult to properly localize the individual data points in the 3D space. This, of course, is an inherent problem with 3D plots, and very likely the authors already explored alternative visualization methods. Still, it might be worth considering a different way of showing the data, perhaps by plotting in 2D and showing the 3rd dimension on a continuous colour scale? The point is that the authors' central argument of different clustering for theta and gamma in this state space is difficult to confirm visually in the current plots.

Response: We have added a supplementary figure showing the requested 2D plots (Supplementary Fig. 3G-H) to the revised manuscript. The labels of each brain region are similar to Fig.3H, I to provide consistent and clear illustration to description in the text. These 2D figures are shown in our response to reviewer #1.

REVIEWERS' COMMENTS:

Reviewer #1 (Remarks to the Author):

The authors have replied convincingly to my comments, and I suggest that the manuscript be published.

Reviewer #2 (Remarks to the Author):

I would like to thank the authors for the revised version of the manuscript. I have no further questions.

Reviewer #3 (Remarks to the Author):

I appreciate that the authors have addressed the raised points very carefully. The new analyses and rewriting efforts substantially improved the manuscript.